



# Offshore Wind Farm Layout Optimization Accounting for Participation to Secondary Reserve Markets

Thuy-hai Nguyen[1], Julian Quick[2], Pierre-Elouan Réthoré[2], Jean-François Toubeau[1], Emmanuel De Jaeger[3], and François Vallée[1]

[1]University of Mons, Electrical Power Engineering Unit, Boulevard Dolez 31, 7000 Mons, Belgium
[2]Technical University of Denmark, Risø National Laboratory for Sustainable Energy, Frederiksborgvej 399, 4000 Roskilde, Denmark
[3]University of Leuven, Department of Mechatronics, Electrical Energy and Dynamic Systems, 2 Place du Levant, 1348 Louvain-la-Neuve, Belgium

**Correspondence:** Thuy-hai Nguyen (thuy-hai.nguyen@umons.ac.be)

**Abstract.** Wind farm layout optimization usually aims at maximizing annual energy production by placing wind turbines in a strategic way to avoid wake losses. However, this might not lead to optimal profits because of the volatility of electricity prices. Moreover, with the growing unpredictability and variability of future power systems due to the increase of renewable electricity production, wind farm operators will have a more important role in balancing the system through participation to reserve markets. This study presents a new formulation for wind farm layout optimization where the objective function aims at maximizing revenues from both day-ahead and reserve markets. It uses stochastic gradient descent for the optimization and probabilistic forecasts for wind power and electricity prices. The new formulation is applied on a test case based on a real-life offshore wind farm in Belgium. An important conclusion is that annual profit is expected to increase in a significant way when accounting for participation to reserve markets, while exhibiting a lower supplied energy production. Moreover, layouts optimized for profit maximization with reserve participation lead to better yearly profits than when considering day-ahead market only in the objective function. Profits are also higher for the new methodology than when using the maximization of annual energy production, widely used in the literature, as objective function.

## 1 Introduction

With the sharp increase of renewable energy sources in modern power systems, balancing electrical load and generation throughout the day is becoming a challenge. In case of real-time imbalance in the system, the Transport System Operator (TSO) needs to activate reserves in order to restore the balance and avoid frequency deviations. In the near future, with a high penetration of weather-dependant electricity generation, the intra-hour variability and randomness will become significant, increasing the need for fast regulation and the value of reserve. Reserve markets, which allow power plant operators to act as Balancing Service Provider (BSP), will be critical for the reliable integration of renewable electricity. Because offshore wind generation capacity is expected to grow steadily in the future, wind farm operators will have an important role in reserve markets and system balancing. Allowing offshore wind farms to participate in the reserve market will be of mutual interest



to TSOs and wind producers. Moreover, it has been proven that variable speed wind turbines in modern wind power plants have intrinsic fast down (virtually at no cost) and ramping up (subject to the availability of wind power) capabilities, which can be effectively used to provide ancillary services (Kayedpour et al., 2024, 2022). To alleviate frequency deviations, the

TSO has several reserve capacities, with different requirements for maximum ramping and activation time. The focus of this work will be on Automatic Frequency Reserve Restoration (aFRR), also called secondary reserve or R2. Indeed, volume needs of secondary reserves are usually higher and are expected to reach even larger values than those for primary reserve in the future (Elia, 2023). Moreover, primary reserve requires an activation and ramping to full capacity within seconds (Perroy et al., 2020), which might be prohibitive within wind farms, where wind and wake effects take time to propagate. Tertiary reserve

is manually activated and is only used to complement and release secondary reserve (e.g., for very extensive imbalances). It must be able to stay active for a long period of time (hours), which could be a challenge for wind farm operators because of the variability of wind. Therefore, secondary reserves seem to be suitable for increasing revenues of wind farms participating to reserve markets (Sumetha-Aksorn et al., 2022; Windvision et al., 2015). Secondary reserves have a fast response time, are used in both directions to restore a frequency of 50 Hz, and remain active as long as necessary. The TSO activates aFRR

automatically by sending a set-point every four seconds and the requested energy is to be activated within 7.5 to 15 minutes in case of selection of the full volume of the aFRR energy bid.

Regarding the participation of wind farms to a Joint day-ahead Energy and Reserve Market (JERM), optimal offering and allocation policies have been investigated, but with the assumption of constant electricity prices (Soares et al., 2017). This does not allow to capture the variation of day-ahead and reserve prices with wind speed and wind direction. A combined energy

and regulation reserve market model has been developed to encourage wind producers to regulate their short-term outputs (Liang et al., 2011), but it assumes that marginal revenues of providing day-ahead energy is always higher than the marginal revenues for upward reserve as well as perfect forecasts of market prices. Provision of reserve by wind power units has been considered for generation capacity expansion (Cañas-Carretón and Carrión, 2020) and simulations were only carried out over 9 representative days of load and generation.

While current wind farms have usually been designed to maximize their power output, future wind farms should be planned and built taking into account the participation to reserve markets. Wind farm layout optimization (WFLO) usually aims at maximizing annual energy production (AEP). It attempts to choose the best placement for turbines, which is equivalent to minimizing wake losses. Indeed, when wind turbines extract mechanical energy from the wind to produce electricity, they cause a reduction of wind speed behind them. Downstream turbines in the wake therefore produce less energy. On a site with specific

wind conditions, WFLO will avoid aligning turbine in the directions of dominant wind. Layout optimization for maximizing AEP has been widely studied in the literature, using gradient-based optimization techniques (Quick et al., 2023; Rodrigues et al., 2023; Park and Law, 2015), gradient-free (Hou et al., 2015; Feng and Shen, 2015; Long et al., 2020), or comparing both (Thomas et al., 2023). The idea behind maximizing AEP is that it will maximize profits for wind farm operators selling energy on the day-ahead energy market (DAEM). However, both objectives might not lead to the same results because of the high

volatility of electricity prices. Producing much energy in periods of low prices will lead to reduced profits. When considering only day-ahead market, if patterns of low and high prices do not match wind direction patterns, optimizing AEP is not the same





as maximizing profit. Indeed, maximizing profit might lead to higher profits while decreasing supplied energy (and thus turbine loads). WFLO for yearly profit has been studied in previous works (Stanley et al., 2021; González et al., 2010; Gonzalez et al., 2012), but wind power was sold only on the day-ahead market. Adding participation to reserve market will also impact results
if day-ahead (DA) and reserve prices do not show the same variations with regard to wind direction.

To the best of the authors' knowledge, this is the first paper that presents a wind farm layout optimization that accounts for the participation to reserve market in the profit objective function. Therefore, the contributions of this paper are threefold.

Firstly, a new formulation for computing the optimal offering, reserve allocation strategy, and subsequent expected profits of
a wind farm participating in both day-ahead and secondary upward reserve markets is developed. It considers the uncertainty in forecasts of wind power, electricity prices and activated reserve volumes. The estimated penalties and balancing costs for failing to provide energy and reserve are also taken into account. The study is conducted for the Belgian system using existing market rules. However, although this system has some peculiarities, the main methodology could be applied in other systems with minor modifications.

Secondly, this new formulation will be used as the objective function of a wind farm layout optimization maximizing yearly profits. Because computing yearly profits at each iteration step of the optimization is too costly, stochastic gradient descent (SGD) is used. This prompts the need to make the profit function differentiable. The gradient of the total profit is estimated for a limited amount of timesteps. This allows to obtain rather accurate results in a reasonable computation time.

Thirdly, the new formulations are applied on a real wind farm, using historical data for wind and electricity prices. When
considering the current built layout, it is shown that operating the wind farm with provision of reserve leads to significantly higher yearly profits than when participating only to DAEM. Then, the new WFLO methodology is applied to optimize the layout while accounting for reserve participation. Yearly profits, supplied energy and AEP of the best optimized layout are compared with regard to the current built configuration of the test wind farm. The optimized layout is also compared with those obtained using the traditional AEP maximization formulation. A profit function only accounting for participation to day-
ahead market is also used for the optimization. The three approaches are compared in terms of expected yearly profits and supplied energy. Finally, generalization to unseen data is studied.

The remainder of the paper is structured as follows. In Section 2, the general formulation for the computation for revenues from both day-ahead and reserve markets is presented, as well as its integration in the wind farm layout optimization problem.
Section 3 details the wind farm optimization test case and briefly analysis historical data from Belgium. Section 4 analyses results and comparisons are made with more traditional wind farm layout optimization formulations. Finally, conclusions and future work are gathered in the last section.





## 2 Methods

One day before real-time delivery (market closure is at noon), for each timestep $t$ of the 24 hours of the next day $k$, a wind
farm operator:

- forecasts available wind power $\hat{P}_{k,t}^{wind,\,avail}$

- decides the total amount of power sold to both day-ahead and reserve markets $P_{k,t}^c$

- decides the amount of reserve capacity to procure to the reserve market $R_{k,t} = \alpha_{k,t} * P_{k,t}^c$

- computes the power to be sold in day-ahead energy market $P_{k,t}^{DA} = P_{k,t}^c - \hat{R}_{k,t}$

The wind farm reserve capacity represents the amount of power that the wind farm holds back from electricity production, to
sell in the reserve market instead of the day-ahead energy market. Based on weather forecasts (and thus wind power forecasts),
a wind farm operator bids its electricity production in the day-ahead market and the reserve capacity in the secondary reserve
market. On the day of delivery, the wind farm must be able to supply both the day-ahead and activated reserve quantities. In
this work, it is assumed that wind farms always prioritize providing reserve (as the wind farm is contractually bound to reserve
this capacity).

The accuracy of weather and thus wind power forecasts is crucial in order to make relevant bids in both markets: under-
estimation leads to lower bids and decreased profits, while overestimating production results in inability to supply contracted
bids, thus incurring financial penalties. Moreover, electricity prices can be highly volatile, and the actual activation of reserve
depends on the system imbalance, which is also fluctuating. Forecast errors on electricity prices and activation volume can
lead to a wrong estimation of expected profit. In this work, we assume that forecast errors follow a gaussian distribution with a
given mean and standard deviation. For each considered timestep, $S$ forecast errors are randomly sampled using a Monte-Carlo
approach.

### 2.1 Wind power forecasts

The forecast of available wind power $\hat{P}_{k,t}^{wind,\,avail}$ depends on (previously) forecasted free-flow wind speed $\hat{u}_{k,t}^{\infty}$ and wind
direction $\hat{\theta}_{k,t}$.

$$\hat{P}_{k,t,s}^{wind,\,avail} = f(\hat{u}_{k,t,s}^{\infty}, \hat{\theta}_{k,t,s})$$

The operator $f(\cdot)$ denotes the conversion of wind data to wind power: it is based on the wind turbines power curve and should
account for wake effects arising within the wind farm. The index $s$ denotes the Monte Carlo sample number related to forecast
error sampling.

The forecasted wind speed $\hat{u}_{k,t}^{\infty}$ is derived from the actual realization of wind speed (normally not known by the wind farm
operator) and a forecast error sampled from a normal distribution.

$$\hat{u}_{k,t,s}^{\infty} = u_{k,t}^{\infty} + \epsilon_{k,t,s}^{u}$$

$$\epsilon_{k,t,s}^{u} \sim N(0, \sigma^u) \tag{1}$$



The same process is used to forecast wind direction

$$\hat{\theta}_{k,t,s}^{\infty} = \theta_{k,t}^{\infty} + \epsilon_{k,t,s}^{\theta}$$
$$\epsilon_{k,t,s}^{\theta} \sim N(0, \sigma^{\theta}) \tag{2}$$

Therefore, forecasts of available wind power can be written as:

$$\hat{P}_{k,t,s}^{avail} = f_P(u_{k,t}^{\infty} + \epsilon_{k,t,s}^{u}, \theta_{k,t}^{\infty} + \epsilon_{k,t,s}^{\theta}) + \epsilon_{k,t,s}^{f_P} \tag{3}$$

$\epsilon_{k,t,s}^{f_P}$ is the modelling error associated with the wind farm model. For wind speed forecasting, literature shows that forecast errors follow a gaussian distribution with mean 0 and standard deviation approximately equal to $15\%$ (ECMWF, 2024). For wind direction, day-ahead forecasts show a root mean squared error of $4.2\,°$ (Chitsazan et al., 2019).

## 2.2 Day-ahead energy market

The day-ahead energy market is a financial market where participants purchase and sell electrical energy at financially binding day-ahead prices for the following day. Electricity is traded at 12h00 for the 24 hours of the next day and the market is cleared based on an auction mechanism, where market price and volume is the intersection point between the demand and supply curves. After the auctions on day-ahead markets are closed, existing shortfalls or surpluses can still be evened out through intra-day trading. However, intra-day market is not considered in this work, as prices are extremely volatile and tend to have similar patterns than imbalance fees. Indeed, market participants are charged with imbalance fees every time they deviate from their nominations. These fees, set on a quarter-hourly basis, aim at ensuring that participants contribute efficiently at balancing the electrical system and reflect the cost related to the activation of additional energy (reserve) by the TSO.

Revenues from the day-ahead market for timestep $t$ of day $k$ can be written as (assuming perfect forecasts):

$$Profit_{k,t}^{DA} = P_{k,t}^{DA} * \lambda_{k,t}^{DA} - \Delta P_{k,t}^{DA} * \lambda_{k,t}^{imb} \tag{4}$$

where $\lambda_{k,t}^{DA}$ is the day-ahead price, $\Delta P_{k,t}^{DA}$ is the contracted power not supplied, and $\lambda_{k,t}^{imb}$ is the imbalance fee.

Day-ahead prices and imbalance penalties need to be forecasted by the wind farm operator before it makes its bid on the market. For the gaussian distribution parameters of day-ahead electricity prices, $\mu$ is approximately 0 and $\sigma$ is around $7\%$ (Lago et al., 2021).

## 2.3 Reserve market for aFRR

Upward regulation is activated in case of negative imbalance in the system (consumption exceeds production), and downward regulation is used for positive imbalance. In this work, we only consider the provision of upward reserve regulation since wind farms are not able to benefit from fuel-saving returns in downward regulation (Toubeau et al., 2020). Indeed, activation of downward reserves can yield both positive (the TSO pays the BSP) and negative prices (the BSP pays the TSO) (Brijs et al., 2015). Negative prices result from producers (e.g., gas-fired power plants) willing to lower their output since their energy is already sold in long-term markets and they can save operating costs: they are usually willing to pay the TSO a small amount.





However, when facing scarcity of downward flexibility, BSP may bid positive activation prices, i.e. being paid for the service, which is the only case where providing downward reserve would be profitable for wind farm operators. It should be noted that this assumption strongly depends on market conditions, but it is suitable for the Belgian case study considered in this work.

Therefore, we will only focus on upward reserve.

Revenues from the aFRR upward market are twofold. BSP earn revenues from the procurement of reserve capacity (through capacity bids), and balancing revenues from the real-time activation of procured reserves. The reserve capacity price $\lambda^{R,c}$ is determined through a pay-as-bid process. We assume that because of the lower production costs for wind generation than conventional power plants, capacity bids from wind farm will be well placed in the merit order and will be chosen first by the TSO.

The reserve activation price $\lambda^{R,a}$ is pay-as-cleared and contracted aFRR energy bids for possible activation on day $k$ have to be submitted by the BSP to the TSO at the latest in day-ahead (day $k$-1). The TSO may activate partially or entirely aFRR energy bids, depending on the negative system imbalance. This process is presented in Fig. 1. The uncertainty in the balancing actions (i.e., the total amount of activated upward reserve) is modeled through scenarios of reserve activation $\kappa^a \in [0, 1]$. Moreover, in case of several market players bidding in the balancing market, we assume an equal distribution of reserve among all market

participants.

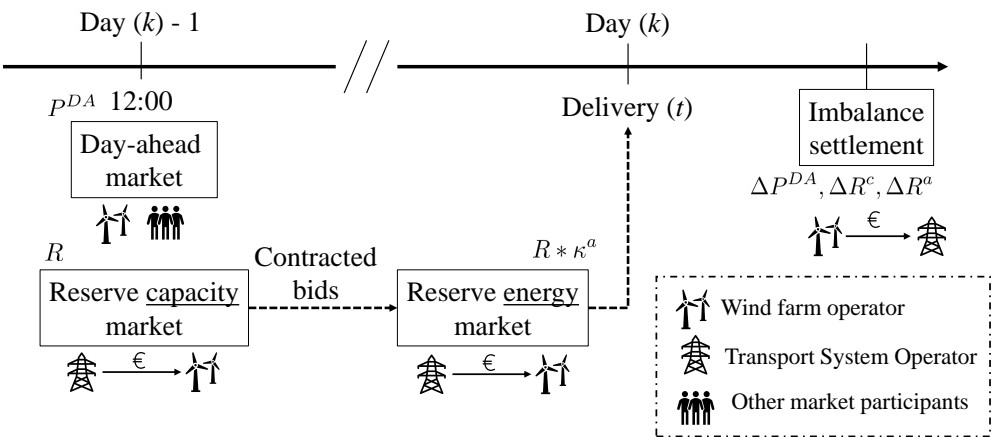

**Figure 1.** Bidding process of an offshore wind farm operator participating in both day-ahead and reserve markets.

Failing to provide the activated reserve requested by the TSO leads to activation penalties that are calculated as follows (Elia, 2022):

$$Penalties_{k,t}^{R,a} = \gamma^a * \frac{Reserve\ discrepancy}{Reserve\ requested} * (Capacity\ remuneration + Activation\ remuneration) \quad (5)$$

where $\gamma^a$ is a penalty multiplier for failing to provide activated reserve. It is set by the TSO and in Belgium, Elia has chosen a value of 1.3 for $\gamma^a$. The reserve discrepancy during activation (contracted reserved not supplied when requested) $\Delta R_{k,t}^a$ is





defined as:

$$\Delta R_{k,t}^a = R_{requested} - R_{supplied}$$

$$\Delta R_{k,t}^a = R_{k,t} * \kappa_{k,t} - \min(R_{k,t} * \kappa_{k,t}, \hat{P}_{k,t}^{wind,\,avail}) \tag{6}$$

In our problem, this translates to this equation:

$$Penalties_{k,t}^{R,a} = 1.3 * \frac{\Delta R_{k,t}^a}{R_{k,t} * \kappa_{k,t}^a} * (R_{k,t} * \lambda_{k,t}^{R,c} + R_{k,t} * \lambda_{k,t}^{R,a} * \kappa_{k,t}^a)$$

$$Penalties_{k,t}^{R,a} = 1.3 * \frac{\Delta R_{k,t}^a}{\kappa_{k,t}^a} * (\lambda_{k,t}^{R,c} + \lambda_{k,t}^{R,a} * \kappa_{k,t}^a) \tag{7}$$

Moreover, Elia controls the availability of the aFRR capacity by performing availability tests. Elia has the right to perform at maximum 12 availability tests on a rolling window of 12 months and each test lasts for 3 quarters of an hour. In case of a failed availability test, the BSP must pay financial penalties.

$$Penalties_{k,t}^{R,c} = \gamma^c * \Delta R_{k,t}^c * \lambda_{k,t}^{R,c} \tag{8}$$

$\Delta R_{k,t}^c$ is the missing reserve capacity during the availability test and $\gamma^c$ is the penalty factor, equal to 0.75 by default. However, in case the penalty concerns a second consecutive failed availability test, $\gamma^c$ is equal to 1.5.

But more importantly, ELIA hinders the possibility of participating to reserve markets by adapting the upper limit of aFRR capacity bids in case of two or more failed consecutive availability tests of the same aFRR capacity product. To account for this technical penalty, we should set a very high penalty price when available power for activation in real-time is lower than reserve capacity bids. This allows to account for this technical constraint in the profit formulation. Therefore, we set $\gamma^c$ to 10.

For a timestep $k, t$ where a wind farm decides to participate to the reserve market, revenues from reserve are computed as follows:

$$Profit_{k,t}^{reserve} = (R_{k,t} * \lambda_{k,t}^{R,c} + R_{k,t} * \lambda_{k,t}^{R,a} * \kappa_{k,t}^a) - (1.3 * \frac{\Delta R_{k,t}^a}{\kappa_{k,t}^a} * (\lambda_{k,t}^{R,c} + \lambda_{k,t}^{R,a} * \kappa_{k,t}^a) + \gamma^c * \Delta R_{k,t}^c * \lambda_{k,t}^{R,c}) \tag{9}$$

Reserve and regulation prices are characterized by higher volatility, lower mean, more frequent price spikes and a more skewed distribution compared to electric energy prices. Thus modelling their behavior is potentially more challenging (Wang et al., 2013).

## 2.4 Profit computation

To summarize, revenues from the participation of both day-ahead and reserve markets over $T$ timesteps of $K$ days can be written as:

$$Profit = \sum_k^K \sum_t^T \mathbb{E}_s \big[ \, (P_{k,t}^{DA} * \hat{\lambda}_{k,t,s}^{DA} + R_{k,t} * \hat{\lambda}_{k,t,s}^{R,c} + R_{k,t} * \hat{\lambda}_{k,t,s}^{R,a} * \hat{\kappa}_{k,t,s}^a)$$

$$- (\Delta P_{k,t,s}^{DA} * \hat{\lambda}_{k,t,s}^{imb} + 1.3 * \frac{\Delta R_{k,t,s}^a}{\hat{\kappa}_{k,t,s}^a} * (\hat{\lambda}_{k,t,s}^{R,c} + \hat{\lambda}_{k,t,s}^{R,a} * \hat{\kappa}_{k,t}^a) + \gamma * \Delta R_{k,t}^c * \hat{\lambda}_{k,t,s}^{R,c}) \, \big] \tag{10}$$



For each timestep $t$ of day $k$, an inner optimization problem gives the optimized total power contracted to the market (day-ahead and reserve), and the percentage of power allocated for the reserve.

$$\underset{\alpha_{k,t},\beta_{k,t}}{Max} \quad \mathbb{E}_s \big[\, P^{DA}_{k,t} * \hat{\lambda}^{DA}_{k,t,s} + R_{k,t} * \hat{\lambda}^{R,c}_{k,t,s} + R_{k,t} * \hat{\lambda}^{R,a}_{k,t,s} * \hat{\kappa}^a_{k,t,s}$$
$$- (\Delta P^{DA}_{k,t,s} * \hat{\lambda}^{imb}_{k,t,s} + 1.3 * \frac{\Delta R^a_{k,t,s}}{\hat{\kappa}^a_{k,t,s}} * (\hat{\lambda}^{R,c}_{k,t,s} + \hat{\lambda}^{R,a}_{k,t,s} * \hat{\kappa}^a_{k,t,s}) + \gamma * \Delta R^c_{k,t,s} * \hat{\lambda}^{R,c}_{k,t,s}) \,\big] \tag{11}$$

with:

$$P^{DA}_{k,t} = (1 - \alpha_{k,t}) * \beta_{k,t} * P^{farm,rated}$$

$$R_{k,t} = \alpha_{k,t} * \beta_{k,t} * P^{farm,rated}$$

$$0 \leq \alpha_{k,t} \leq 1 \; \forall k, t$$

$$0 \leq \beta_{k,t} \leq 1 \; \forall k, t \tag{12}$$

$$R_{k,t} \in [0, R_{max}] \tag{13}$$

The agreed upon power schedules, $P^{\mathrm{DA}}_{k,t}$ and $R_{k,t}$ are the true design variables in this problem. The total contracted power in reserve and day-ahead markets cannot exceed the wind farm installed capacity, which is translated with constraints (12). Moreover, reserve bids are limited to a maximum value $R_{max}$, ensured by constraint (13). Indeed, according to Elia rules for BSP participating to aFRR markets, each bid should not exceed 50 MW per delivery point. Furthermore, aFRR requirements for the Belgian power system was 117 MW in 2023 (total power contracted by Elia with BSPs), which sets an absolute value as well. For each timestep, the wind farm operator chooses to contract $P^c_{k,t}$, the total contracted power, to the JERM. This quantity is optimized through the $\beta_{k,t}$ variable. The allocation of this contracted power to the day-ahead and reserve markets is then given with $\alpha_{k,t}$. As a reminder, in case of missing power (available power lower than power contracted in the JERM, i.e. $\Delta P^{DA}_{k,t,s}$ and/or $\Delta R^a_{k,t,s} \geq 0$), the supply of activated reserve will always be prioritized, regardless of imbalance prices.

This optimal allocation of day-ahead and reserve power is similar to the flexible stochastic formulation available in the literature (Soares et al., 2017). This approach is characterized by its total freedom to choose the energy and reserve share in each stage of the problem; i.e. the wind farm can take advantage of the intermediate information about wind power production, thereby reducing the penalties at the balancing stage. This means that the operator can adjust the share of energy and reserve in the balancing stage in line with the expected power production in each scenario $s$. Optimal values of $\alpha_{k,t}$ and $\beta_{k,t}$ can be found with a combinatorial exploration (since their range is limited and granularity do not have to be very high, as power bids are submitted by steps of 1 MW).

## 2.5 Layout optimization

Taking into account uncertainty on wind (thus wind power) and price forecasts, we can write the optimization problem:



$$
\underset{\boldsymbol{x},\boldsymbol{y}}{Max} \sum_k^K \sum_t^T \mathbb{E}_s \left[ \, P_{k,t}^{DA} * \hat{\lambda}_{k,t,s}^{DA} + R_{k,t} * \hat{\lambda}_{k,t,s}^{R,c} + R_{k,t} * \hat{\lambda}_{k,t,s}^{R,a} * \hat{\kappa}_{k,t,s}^{a} \right.
$$

$$
\left. - (\Delta P_{k,t,s}^{DA} * \hat{\lambda}_{k,t,s}^{imb} + 1.3 * \frac{\Delta R_{k,t,s}^{a}}{\hat{\kappa}_{k,t,s}^{a}} * (\hat{\lambda}_{k,t,s}^{R,c} + \hat{\lambda}_{k,t,s}^{R,a} * \hat{\kappa}_{k,t,s}^{a}) + \gamma * \Delta R_{k,t,s}^{c} * \hat{\lambda}_{k,t,s}^{R,c}) \, \right]
$$

subject to

$$
\sqrt{(x_i - x_j)^2 + (y_i - y_j)^2} \leq d_{min} \; \forall \, i, j > i \tag{14}
$$

$$
x^l \leq x_i \leq x^u \; \forall i
$$
$$
y^l \leq y_i \leq y^u \; \forall i \tag{15}
$$

The design variables are $\boldsymbol{x}$ and $\boldsymbol{y}$, the vectors of x- and y-coordinates of wind turbines. Constraint (14) ensures a minimum spacing $d_{min}$ between adjacent turbines while (15) keeps turbines from being outside the farm boundaries ($[x^l$-$x^u]$, $[y^l$-$y^u]$). The objective function aims at maximizing the total profit over $T$ timesteps of $K$ days. The total power contracted in the JERM, the allocation of reserve and the distribution of potential missing power is determined from Eq. (11). The complete methodology is summarized in Fig. 2.

The optimization is carried out using stochastic gradient descent, which is an iterative method for optimizing a differentiable objective function. It replaces the actual gradient (calculated from the entire data set) by an estimate (calculated from a randomly selected subset of the data). Therefore, the algorithm follows the mean gradient by a specified distance, which is equivalent to optimizing the expected value of the objective function (Quick et al., 2023). This reduces the very high computational burden in high-dimensional optimization problems, achieving faster iterations but at the cost of a lower convergence rate. The inner optimization for the optimal bidding strategy in the JERM, defined by Eq. (11), is solved through a combinatorial exploration, which makes it differentiable and thus compatible with the SGD algorithm. Because computing the total profit for a year (365 days * 96 quarters of an hour, i.e., 35,040 timesteps) at each iteration would bee too costly, SGD is particularly relevant for our proposed WFLO formulation.

## 3   Test case

We use data from Northwind, a Belgian offshore wind farm situated 38 km from the coast, in the North Sea, within the first Belgian offshore cluster. Northwind consists of 72 Vestas turbines, for a total installed capacity of 216 MW. Each turbine has a rotor diameter of 112 m, a hub height of 71 m, and the rated power is 3.075 MW. The layout of this wind farm can be seen in Fig. 3.

Before optimizing the layout, expected yearly profits and supplied energy are computed for the current built layout (further referred to as the base layout), for different modes of operation (with and without reserve). Then the maximum amount





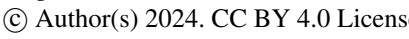

Figure 2. Methodology for layout optimization accounting for reserve participation

of power that can be allocated to reserve $R_{max}$ is set to different values. First, Elia has set a limit of 50 MW per delivery point in its current BSP agreement. Then, the required volume of aFRR reserves that Elia should ensure throughout the year was 117 MW for 2023. However, with the growing penetration of renewable energies foreseen in the future, one can expect that this requirement will increase as well. Indeed, power systems will become highly weather-dependant, thus more prone to

245   variability and unpredictability. We therefore set two more values for the maximum allocated reserve: Elia's total aFRR needs (approximately equal to half of the wind farm rated capacity, if the operator wants to keep a part of available wind power for other markets), and the full farm capacity.





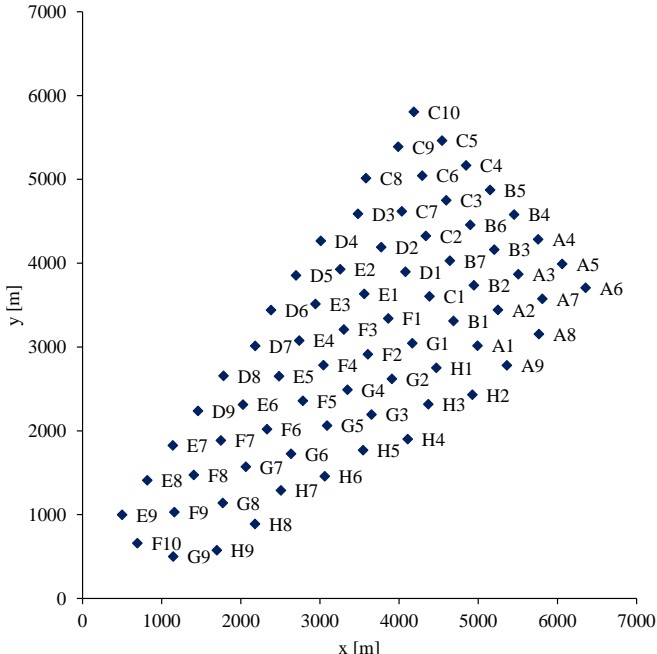

**Figure 3.** Layout of Northwind offshore wind farm

WFLO is then carried out with the new formulation for the objective function that maximizes profit from both day-ahead and
250    reserve markets. Historical data from 2023 in Belgium (see section 3.1) are used during the optimization process. To assess the influence of accounting for reserve in the WFLO process, optimization with day-ahead market only is also simulated. Then, to compare with state-of-the-art WFLO formulations, the layout will also be optimized with the objective of maximizing AEP. Results will be compared in terms of expected yearly profits and yearly production. Moreover, it is important that the optimal layouts are not only relevant for the data used in the optimization process. Therefore, yearly profits are also computed for
255    unseen data, i.e. historical data from another year (2024).

The new objective function for WFLO has been integrated into the TOPFARM framework (Riva et al., 2024), comprising the SGD optimizer. Profit and AEP gradients are computed using automatic differentiation. Turbine powers are obtained from Pywake simulations (Pedersen et al., 2023), using the Bastankhah Gaussian wake model for velocity deficits, the Crespo-
260    Hernandez turbulence model for added turbulence and linear superposition. The minimum turbine spacing $d_{min}$ (constraint of Eq. (14)) is set to 2 rotor diameters. The SGD optimizations are carried out using the following parameters: the initial learning rate is one rotor diameter, the maximum number of iterations is 2000, and the initial value for constraint aggregation multiplier is 0.1. We use several values of $K * T$ (numbers of samples for every SGD iteration) when optimizing for profits and AEP. To obtain statistically significant results, each case of SGD optimization is run using 5 different initial random starting conditions.





## 3.1 Analysis of historical data in Belgium

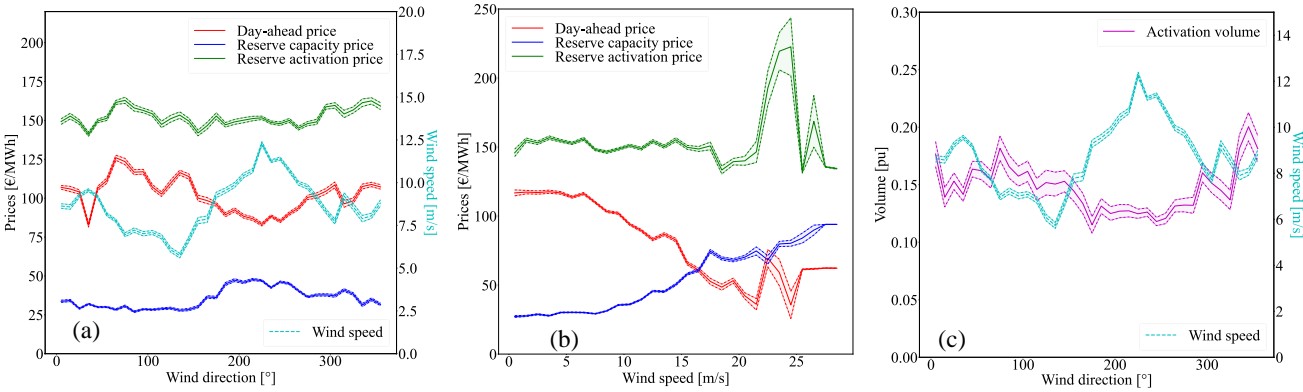

**Figure 4.** Mean electricity prices with regard to (a) wind direction and (b) wind speed in 2023. (c) Mean normalized activated volumes of reserve with regard to wind direction.

To help better understand the motivations of this work, we analyse historical data of wind, electricity prices and activated reserve volume for the year 2023 in Belgium. Data of wind speed and wind direction at the location of the offshore wind farms in the Belgian North Sea have been gathered from ERA5 database. Electricity prices for the day-ahead market were available on the European Network of Transmission System Operators (ENTSO-E) transparency platform (ENTSO-E, 2024). Prices for reserve capacity and reserve activation, as well as activated upward aFRR reserve volumes are were provided by Elia, the Belgian TSO (Elia, 2024). From Fig. 4, we can study the variations of price with regard to wind direction and wind speed, and the mean activated reserve per wind sector. It can be seen in Fig. 4(a) that mean day-ahead prices do not follow the same pattern as mean reserve capacity and activation prices with regard to wind direction. Indeed, mean day-ahead prices show a lower mean value for the wind sector centered around 230 °. This wind sector corresponds to the direction of dominant wind in this area of the North Sea (direction with most occurrences), as it can be seen in Fig. 5. This leads to a discrepancy between maximizing profits and energy production. Indeed, when prices are not considered, WFLO will try to avoid aligning turbines in the dominant wind direction. However, since prices tend to be lower in that wind section, it might be more profitable to avoid wake losses in other directions, where prices are higher. Mean reserve capacity prices, on the other hand, tend to be higher in that wind direction sector, while reserve activation prices do not show a significant increase or decrease. This means that accounting for participation to reserve will affect the optimization results, as day-ahead and reserve prices have different patterns with regard to wind direction. Fig. 4(c) shows the volumes of activated reserve normalized by the maximum activated volume for aFRR upward reserve (117 MW in 2023). We can see that mean activated volumes tend to be lower in the direction of dominant wind. Another interesting analysis can be made in Fig. 4(b), which displays the mean electricity prices with regard to wind speed. One can see that day-ahead prices tend to decrease with higher wind speeds, while it is the opposite for reserve capacity prices. One reason that explains this reduction of day-ahead price with increasing wind speed is the high penetration of





offshore wind generation in the Belgian power system (10% of yearly consumption produced by offshore wind farms). Because wind energy has lower production costs than conventional thermal power plants, a high production of electricity through wind turbines can lead to lower prices in the day-ahead market. Reserve activation prices remain constant until approximately 20 m/s, but show a sharp increase around 25 m/s, which is the cut-off wind speed of most Belgian offshore wind turbines. This is the

290 limit at which turbines are shut down to prevent mechanical damage, and the farm output goes from the rated power to 0. Therefore, a small prediction error in wind speed can lead to a tremendous need of reserve.

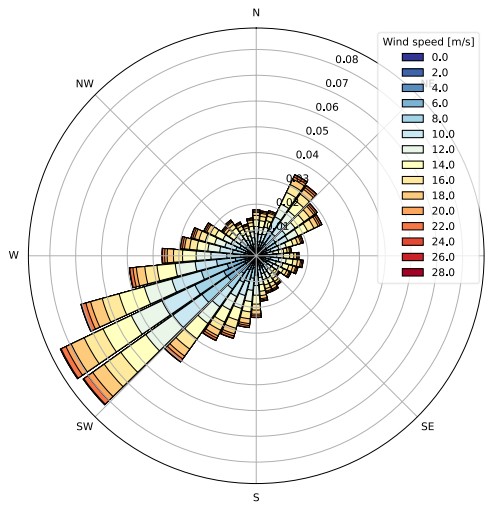

**Figure 5.** Wind rose at the location of Belgian offshore wind farms for 2023, from ERA5 data (latitude: 51.5°N, longitude: 2.75°E)

## 4 Results and discussions

### 4.1 Operating the current built layout with reserve participation

Before optimizing the layout of the Northwind wind farm, yearly profits are computed for the base layout (currently built)

295 using historical data from 2023. Three modes of operation are considered:

- Producing as much wind power as possible (referred to as prod. max. operation in results tables). Energy bids are not risk-based as they only rely on forecasts of available power, regardless of market conditions. This is the most simple operation as the operator does not need to derate the turbines in case of unfavorable market conjuncture.

- Wind power is only sold on the day-ahead market but energy bids are made based on forecasts of available wind power,

300 day-ahead prices and imbalance penalties (referred to as DAEM optimized operation in results tables)

- Wind power is sold on JERM (provision of reserve)





The optimal allocation of day-ahead and reserve power on JERM is solved with Eq. (11) for every quarter of an hour of the year, and the expected profits are summed over 35,040 timesteps. The maximum value allowed for reserve bids is first set to 50 MW. The optimized operation on DAEM only uses the same formulation but with $R_{k,t} = 0 \; \forall k, \forall t$. This allows to assess the impact of participating to the upward secondary reserve market.

The expected supplied energy is the yearly production of the farm actually injected to the grid. For the JERM case, it encompasses both the energy sold on the day-ahead energy market and the activated reserve. Note that electrical losses and downtime due to maintenance and failures are not taken into account. AEP$_{theory}$ is the *theoretical* yearly production of the farm, computed solely by converting data of wind speed and wind direction to potential wind power. It does not include any forecasting errors. Expected yearly profits and supplied energy are reported as $\mu \pm \frac{\sigma}{\sqrt{S}}$, where the standard deviation relates to forecast uncertainty (sampling of $S$ forecast errors).

**Table 1.** Expected yearly profits and supplied energy in 2023 for the initial base layout of Northwind, operated for maximizing production on DAEM (a;d), maximizing profits on DAEM only (b;e), and maximizing profits on JERM (c;f, with maximum reserve bids $R_{max}$=50 MW). Results reported as $\mu \pm \frac{\sigma}{\sqrt{S}}$, where $\mu$ and $\sigma$ relate to forecast uncertainty.

| | Base layout | |
|---|---|---|
| Expected yearly profits | | |
| (a) On DAEM only (operation for prod. max.) | 60.6808 ± 0.0079 M€ | |
| (b) On DAEM only (operated for optimized profits) | 64.6285 ± 0.0079 M€ | ▲6.51% w.r.t. (a) |
| (c) On JERM (operated with reserve) | 69.6735 ± 0.0084 M€ | ▲7.81% w.r.t. (b), ▲14.82% w.r.t. (a) |
| Expected energy supplied | | |
| (d) On DAEM only (operation for prod. max.) | 813.48 ± 0.04 GWh | |
| (e) On DAEM only (operated for optimized profits) | 777.14 ± 0.05 GWh | ▼4.47% w.r.t. (d) |
| (f) On JERM (operated with reserve) | 717.64 ± 0.05 GWh | ▼7.66% w.r.t. (e), ▼11.78% w.r.t. (d) |
| AEP$_{theory}$ | 919.78 GWh | |

It can be seen in Table 1 that operating Northwind for maximizing production leads to the lowest profits. Indeed, making energy bids on DAEM only for profits maximization increases expected yearly profits by 6.51%. This can be explained by two factors. First, producing much wind power when day-ahead prices are negative is detrimental, but timesteps with such prices only occur 2.52% of the time in data from 2023. Then, forecasting prices allows to adopt a risk-aware approach, i.e., bidding more when imbalance penalty prices are expected to be close to day-ahead prices (the risk is acceptable), and bidding less than the forecasted available power in case of very high imbalance prices. This confirmed in Table 2, which shows the profits breakdown between positive profits and imbalance penalties. Total imbalance penalties are higher when operating for profit maximization, but the significant increase in positive profits allows to compensate for the penalties losses. Regarding supplied energy, maximizing production obviously leads to more wind power injected to the grid. The reduction of supplied energy of 4.47% when maximizing DAEM profits is very interesting, because it could lead to lower load constraints on wind turbines, thus extending turbines lifetime. However, this aspect needs to be further investigated.



Supplying secondary upward reserve (operate the wind farm on JERM) increases expected yearly profits by 7.81%, while the supplied energy is decreased by 7.66%. Indeed, bidding a given amount of power in the reserve capacity and energy markets

does not mean that this power will be entirely supplied. If the system negative imbalance is not too severe, only a fraction of contracted reserves is actually activated by the TSO. However, the wind farm operator still earns profits by making this power available to restore balance in the system. This is particularly profitable when day-prices are very low. It can be seen in Table 2 that while positive profits on DAEM are quite lower when providing reserve (earnings are "transferred" to the reserve markets), imbalance penalties do not decrease significantly. This is inherent to our formulation, because in case of imbalance (available

wind power is lower than total power bidded on both DAEM and reserve markets), priority is given to the reserve provision.

**Table 2.** Breakdown of expected yearly profits in 2023 for the initial base layout of Northwind, operated for maximizing production on DAEM (a), maximizing profits on DAEM only (b), and maximizing profits on JERM (c, with maximum reserve bids $R_{max}$=50 MW).

|  | Positive profits on DAEM | Imbalance penalties on DAEM | Reserve profits | Reserve penalties |
|---|---|---|---|---|
| (a) On DAEM only (operation for prod. max.) | 75.7382 M€ | 15.0574 M€ | / | / |
| (b) On DAEM only (operated for optimized profits) | 87.2550 M€ | 22.6264 M€ | / | / |
| (c) On JERM (operated with reserve) | 82.5963 M€ | 22.5231 M€ | 9.9439 M€ | 0.3435 M€ |

**Sensitivity to reserve limit $R_{max}$**

Currently, the maximum value per delivery point of reserve capacity bids established by Elia is 50 MW. Moreover, the static volume need of aFRR reserves for the Belgian power system in 2023 was set at 117 MW. However, as stated before, in future weather-dominated power systems, the need for frequency regulation, including aFRR, will increase. Therefore, the wind farm

is operated considering for 2 other values of $R_{max}$: 117 MW (approximately 1/2 of rated capacity, in case the operator always wants to keep a part of available wind power for other markets), and 221.4 MW (full farm capacity).

With $R_{max}$=117 MW, it can be observed in Table 3 that for the base layout, expected yearly profits increase by 15.15% when the wind farm offers aFRR services, while supplied energy drops by 14.6%. Compared to the previous case ($R_{max}$=50 MW),

doubling the allowed maximum value for reserve capacity bids leads to a profit improvement also multiplied by 2 (15.15% against 7.81% previously). Regarding yearly supplied energy (on DAEM and activated reserve), it decreases when $R_{max}$ is increased ($\approx$ 664 GWh against $\approx$ 718 GWh with $R_{max}$=50 MW). Indeed, allowing for higher reserve bids enables wind farms to participate more in frequency reserve services. But if reserve energy bids are not entirely activated by the TSO, less energy is supplied, while profits are increased.

With the full wind farm capacity (221.4 MW) as $R_{max}$, expected yearly profits on JERM in 2023 for the base layout are even higher. However, even though $R_{max}$ is doubled compared to the previous case, profit increments are not multiplied by 2 this time. Indeed, the improvement is at 20.60%, against 15.15% when $R_{max}$ was set at 117 MW. This shows a flattening of profits augmentation, and wind farm operators might want to avoid allocating all available power to the reserve market. Indeed,





**Table 3.** Expected yearly profits and supplied energy in 2023 for the initial base layout of Northwind, operated for maximizing production on DAEM (a;f), maximizing profits on DAEM only (b;g), and maximizing profits on JERM (c-e;h-j, for different limits on reserve participation, $R_{max}$). Results reported as $\mu \pm \frac{\sigma}{\sqrt{S}}$, where $\mu$ and $\sigma$ relate to forecast uncertainty.

|  | Base layout | Comparison |
|---|---|---|
| Expected yearly profits |  |  |
| (a) On DAEM only (operation for prod. max.) | 60.6808 $\pm$ 0.0079 M€ |  |
| (b) On DAEM only (operated for optimized profits) | 64.6285 $\pm$ 0.0079 M€ |  |
| (c) On JERM (operated with reserve, $R_{max}$=50MW) | 69.6735 $\pm$ 0.0084 M€ | ▲7.81% w.r.t. (b) |
| (d) On JERM (operated with reserve, $R_{max}$=117MW) | 74.4216 $\pm$ 0.0093 M€ | ▲15.15% w.r.t. (b) |
| (e) On JERM (operated with reserve, $R_{max}$=221.4MW) | 77.9445 $\pm$ 0.0106 M€ | ▲20.60% w.r.t. (b) |
| Expected energy supplied |  |  |
| (f) On DAEM only (operation for prod. max.) | 813.48 $\pm$ 0.04 GWh |  |
| (g) On DAEM only (operated for optimized profits) | 777.14 $\pm$ 0.05 GWh |  |
| (h) On JERM (operated with reserve, $R_{max}$=50MW) | 717.64 $\pm$ 0.04 GWh | ▼7.66% w.r.t. (g) |
| (i) On JERM (operated with reserve, $R_{max}$=117MW) | 663.61 $\pm$ 0.05 GWh | ▼14.61% w.r.t. (g) |
| (j) On JERM (operated with reserve, $R_{max}$=221.4MW) | 626.44 $\pm$ 0.04 GWh | ▼19.39% w.r.t. (g) |
| $\text{AEP}_{theory}$ | 919.78 GWh |  |

because of potential forecast errors, there is a significant risk to bid in only one market, and operators could want to keep a part of available wind power for other markets (or even a security margin to avoid penalties when contracted power cannot be entirely supplied).

### 4.2 Optimized layout accounting for reserve participation

WFLO is carried out with the objective of maximizing profits on JERM. SGD optimizations are performed for different values of Monte-Carlo samples ($K * T$) and several initial conditions. Reported results are those obtained with the best optimized layout out of all simulations, i.e. the one leading to the highest expected yearly profits on JERM.

As it can be seen in Table 4, the best optimized layout leads to an increase of yearly profits on JERM by 3.10%, as well as 3.17% more supplied energy. This augmentation of production could be explained by two reasons. On one hand, a better placement of wind turbines to avoid wake losses leads to an improved electricity production in general. On the other hand, this increase of power output coincides with wind directions related to higher electricity prices, which boosts profits. Considering that the average lifespan of an offshore wind farm is approximately 20 years (Topham and McMillan, 2017), a profit increased by 2.16 M€ per year leads to a significant improvement in the wind farm profitability: more than 40 M€ over the farm lifetime. Note that these numbers cannot be directly generalized for other electricity pools, but since most European electricity markets have a similar structure, applying this methodology is also expected to result in higher yearly profits for layouts optimized for



**Table 4.** Expected yearly profits and supplied energy in 2023 for the best layout optimized on JERM, operated for maximizing production on DAEM (a;d), maximizing profits on DAEM only (b;e), and maximizing profits on JERM (c;f, with maximum reserve bids $R_{max}$=50 MW). Results reported as $\mu \pm \frac{\sigma}{\sqrt{S}}$, where $\mu$ and $\sigma$ relate to forecast uncertainty.

| | Base | Layout optimized on JERM |
|---|---|---|
| Expected yearly profits | | |
| (a) On DAEM only (operation for prod. max.) [M€] | $60.6808 \pm 0.0079$ | $62.8504 \pm 0.0076$ (▲3.58% w.r.t. base) |
| (b) On DAEM only (operated for optimized profits) [M€] | $64.6285 \pm 0.0079$ | $66.7564 \pm 0.0081$ (▲3.29% w.r.t. base) |
| (c) On JERM (operated with reserve) [M€] | $69.6735 \pm 0.0084$ | $71.8338 \pm 0.0083$ (▲3.10% w.r.t. base) |
| Expected energy supplied | | |
| (d) On DAEM only (operation for prod. max.) [GWh] | $813.48 \pm 0.04$ | $836.44 \pm 0.04$ (▲2.82% w.r.t. base) |
| (e) On DAEM only (operated for optimized profits) [GWh] | $777.62 \pm 0.05$ | $800.40 \pm 0.05$ (▲2.99% w.r.t. base) |
| (f) On JERM (operated with reserve) [GWh] | $717.98 \pm 0.04$ | $740.42 \pm 0.05$ (▲3.17% w.r.t. base) |
| $AEP_{theory}$ [GWh] | $919.78$ | $942.71$ (▲2.49% w.r.t. base) |

profit maximization with reserve participation.

The best optimized layout is plotted in Fig. 6 and compared to the current layout of Northwind. Turbines that were on the farm boundaries in the base layout kept their position on the outer limits (even though turbines had random positions in the starting initial conditions of the optimization). However, inner turbines positions have been significantly modified compared to
the base layout. Indeed, while the structure of rows has been approximately maintained, it can be observed that more turbines are placed together on a row, while consecutive rows are more distant from one another and are not parallel (which was the case for all rows in the base layout). A few turbines have a more irregular position (in-between rows).

Another interesting characteristic to compare between the optimized and base layouts is the power rose. It shows the power
output of the wind farm with regard to wind direction, for a given wind speed. In Fig. 7, the power is normalized by the wind farm rated capacity (for Northwind, 221.4 MW). It allows to identify the wind directions leading to higher wake losses.

The power rose of Fig. 7(a) shows that the base layout exhibits many power drops, with wake losses being the most severe for directions of 135° and 315° (0° corresponds to wind blowing from the North, then clockwise counting). This pattern is inherent to regular layouts, where turbines are placed in rows equidistant from each other and wake losses are at their maximum
when most turbines are aligned with wind direction. In Fig. 7(b), wake losses are still prevalent for wind directions of 135° and 315°, but they are less severe, and power drops are smoothed for other directions. Indeed, while still located on rows, turbines from different rows are more distant, allowing wind speed to recover between consecutive rows. However, it should be noted that a more irregular turbine placement can lead to higher installation costs and fatigue loading (Sickler et al., 2023).





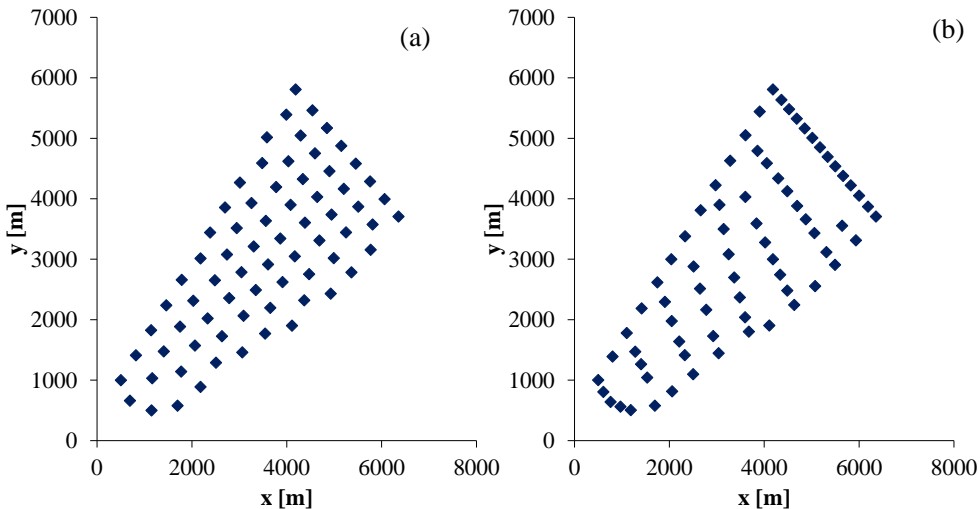

**Figure 6.** (a) Base layout of Northwind offshore wind farm, (b) Best layout optimized for profit maximization with participation to reserve market.

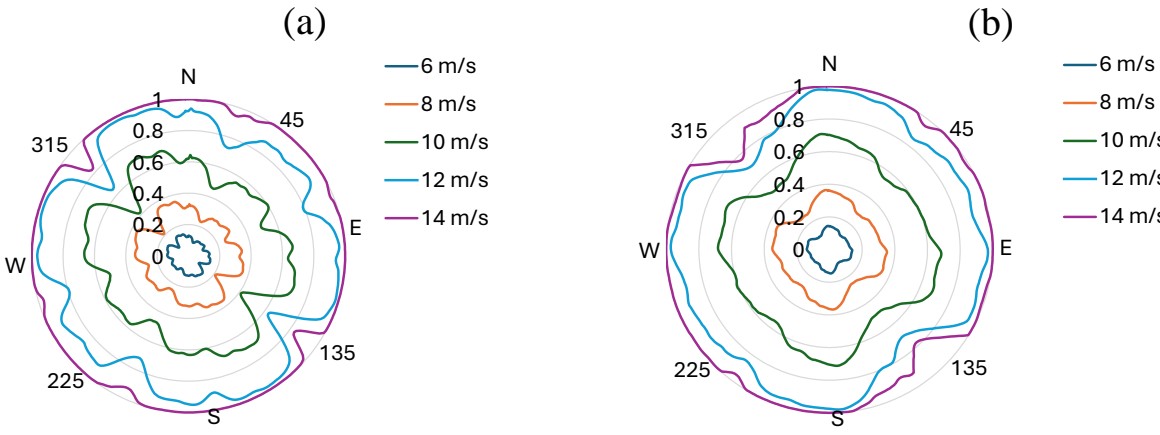

**Figure 7.** Power roses of Northwind (a) Base layout, (b) Layout optimized for profit maximization in JERM.

### 4.3 Comparison with layout optimized without reserve

To assess the impact of including participation to reserve in the layout optimization process, the objective function has been modified as to only include profits from the day-ahead market (i.e., setting $R_{max} = 0$ MW), referenced as the DAEM case. Like before, SGD optimizations are performed for different values of Monte-Carlo samples ($K * T$) and several initial conditions. For every obtained layout optimized without consideration of reserves in the objective function, expected profits on DAEM





only (i.e., wind farm operated without reserve) are computed. The optimized layouts for DAEM are also operated with reserve
in order to compare yearly profits on JERM with those computed for the layouts optimized with reserve participation.

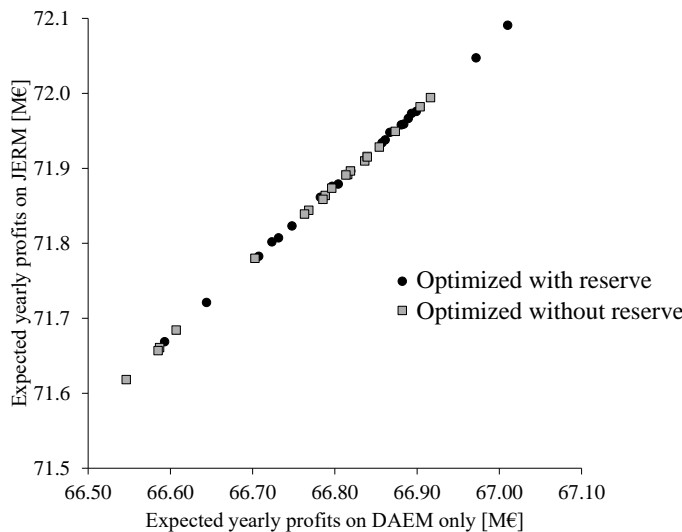

**Figure 8.** Expected yearly profits on JERM plotted versus expected yearly profits on DAEM only (wind farm operated without reserve).
Each point corresponds to one optimized layout, with the black circles representing layouts optimized with reserve, and the grey squares are
for layouts optimized without reserve.

Fig. 8 shows the expected yearly profits on JERM in function of expected yearly profits on DAEM only for the layouts
optimized for profits maximization with and without reserve participation. The highest profits on JERM are obtained for
layouts optimized with reserve: this highlights the importance of accounting for participation of wind farms to reserve markets
in the layout optimization process. Moreover, it can be seen that both yearly profits are linearly linked, i.e., higher profits
on DAEM only leads to higher profits on JERM. Surprisingly, the best total yearly profits on DAEM only are obtained for
cases optimized with reserve. One reason that could explain this is that during the optimization without reserve, the wind farm
can only participate to one market (the DAEM). If day-ahead prices are very low or negative, wind farm will not bid on the
DAEM, resulting in no profit, thus leading to a zero gradient and the solution space being less explored. This means that even
if reserve market rules change dramatically, causing the wind farm to be unable to participate in the reserve market, operating
the optimized layout with reserve on DAEM only would still be profitable.

### 4.4  Comparison with optimized layout for AEP maximization

We compare our novel formulation for WFLO with the objective function widely used in the current literature, i.e., AEP
maximization. For the latter, wind speeds and wind directions from the 2023 historical data are used during the optimization
process. To benchmark the performance of our methodology, expected yearly profits on JERM are computed with reserve





participation, for the layouts optimized for AEP maximization. It is worth reminding that while the total energy supplied indicates the actual electricity sold (or activated, in case of reserve provision) and injected to the grid, AEP gives the theoretical energy that could be supplied by the wind farm given the wind conditions, regardless of prices and erorrs on wind power forecasts.

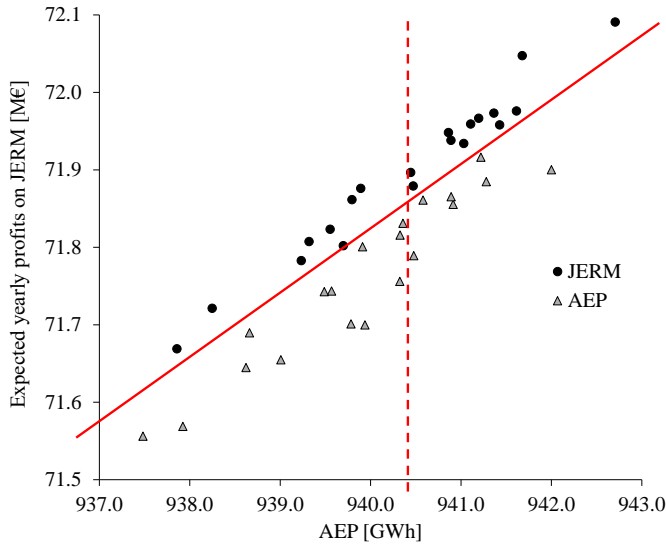

**Figure 9.** Expected yearly profits on JERM plotted versus AEP. Each point corresponds to one optimized layout, with the black circles representing layouts optimized with reserve, and the grey triangles are for layouts optimized for AEP maximization.

Fig. 9 shows the expected yearly profits on JERM in function of AEP for the layouts optimized for profits maximization
with reserve and AEP maximization. The uppermost black circle on the right represents the optimized layout giving the highest profits on JERM but also the highest AEP. It may seem confusing at first that the best AEP is not obtained for a layout optimized for AEP maximization. A plausible explanation is that the profit function has better gradients than the AEP objective function, allowing to avoid local minima. Note that this layout corresponds to the best layout optimized for profits maximization with reserve for which results are given in Table 4.

Another interesting observation is that yearly profits on JERM are generally higher for layouts optimized for profits maximization with reserve: if a diagonal is drawn in the scatterplot, all triangles are located below that line compared to the circles. And if a vertical line is plotted for a given AEP, black circles are always located above the triangles. In other words, for the same level of AEP, the layouts optimized for reserve lead to higher profits on JERM than the ones obtained for AEP maximization. The explanation for those significant economic losses is that the objective function with AEP aims at maximizing the
power output of wind farm regardless of electricity prices. It usually avoids wake losses for the directions of dominant wind. However, if low or even negative prices are associated with those directions, then profits will not increase. Moreover, besides profits, it is not beneficial for the grid that wind farms produce a lot of electricity when prices are quite low. Indeed, for power





systems with a high penetration of renewable energies, especially wind, low or even negative prices may correspond to periods
of overproduction, i.e., generation exceeds consumption. In that case, wind turbines might have to be shut down to curtail wind
energy and restore balance in the system. This spillage of renewable energy is of course not desirable and it is much more
relevant to optimize wind farm layouts so that they produce more energy during times of scarcity (usually associated with
higher prices).

## 4.5   Generalization to unseen future data

In Sections 4.2 to 4.1, historical data from 2023 were used during the SGD optimizations, as well as for the computation of
expected yearly profits for the optimized layouts. In this section, revenues and supplied energy will be assessed with historical
data from 2024, i.e., data unseen during the optimization process. Indeed, it is valuable to have optimized layouts that also
yield improved profits for future years.

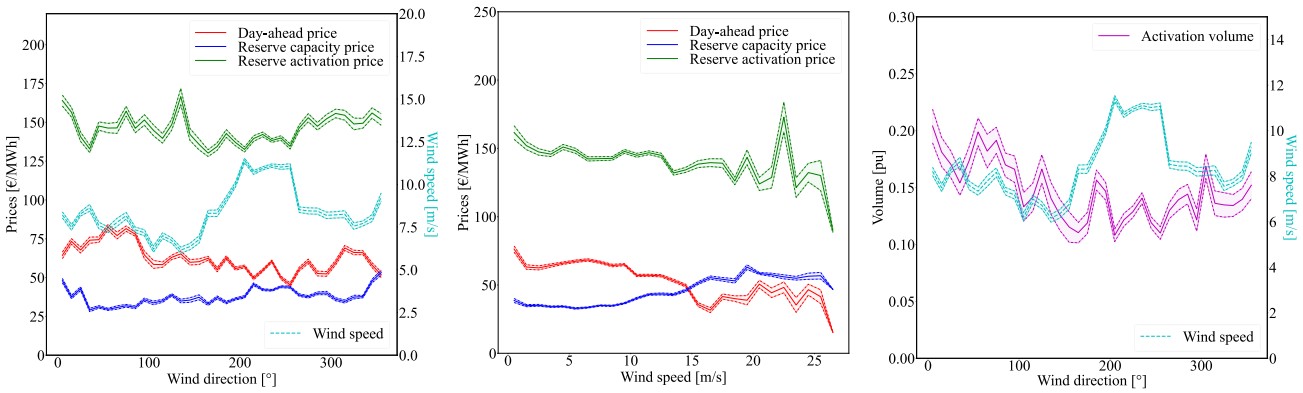

**Figure 10.** Mean electricity prices with regard to (a) wind direction and (b) wind speed in 2024. (c) Mean normalized activated volumes of
reserve with regard to wind direction.

First, wind data and electricity prices from January to July 2024 are analyzed. The wind rose of 2024, plotted in Fig. 11,
shows patterns comparable with 2023: dominant wind directions are mostly South-Westerly. More wind blowing from the
North-East was visible in Fig. 5, which is not the case here. It can be seen in Fig. 10(a) that day-ahead prices, similar to 2023,
have lower values for the direction of dominant winds, although this is less noticeable than in 2023. Moreover, day-ahead prices
in 2024 have overall lower mean values than in 2023. Reserve capacity prices do not vary much with wind direction, while
activation prices show more variability but no significant drop for the directions of dominant wind. The overall mean values
are in the same order of magnitude than for the previous year, and we observe again a sharp increase in reserve activation
prices between 20 and 25 m/s. However, this peak is less pronounced, with a mean peak value under 200€/MWh while it
reached almost 250€/MWh in 2023. This could be explained by less sudden high wind events (e.g., storms), a smoother farm
cut-out or a better anticipation by the TSO. Normalized activation volumes exhibit lower values for directions of dominant
wind. Therefore, while wind, prices and activated reserve volume of 2024 share some similarities with data from 2023, they





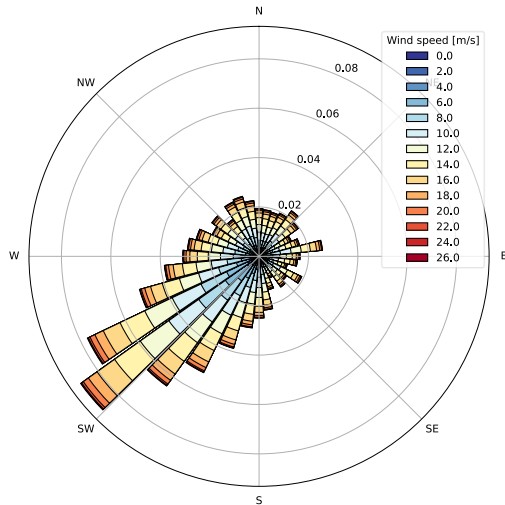

**Figure 11.** Wind rose at the location of Belgian offshore wind farms for 2024, from ERA5 data (latitude: 51.5°N, longitude: 2.75°E)

also exhibit noticeable differences. They are thus relevant to test the validity of the optimized layouts on unseen data. The case
presented here uses $R_{max}$=50 MW when the wind farm is operated with reserve.

**Table 5.** Expected yearly profits and supplied energy in 2024 for the base and best layout optimized on JERM with 2023 data, both operated
for maximizing production on DAEM (a;d), maximizing profits on DAEM only (b;e), and maximizing profits on JERM (c;f, with maximum
reserve bids $R_{max}$=50 MW). Results reported as $\mu \pm \frac{\sigma}{\sqrt{S}}$, where $\mu$ and $\sigma$ relate to forecast uncertainty.

| | Base | Layout optimized on JERM |
|---|---|---|
| Expected yearly profits | | |
| (a) On DAEM only (operation for prod. max.) [M€] | $18.8916 \pm 0.0053$ | $19.6934 \pm 0.0049$ (▲4.24% w.r.t. base) |
| (b) On DAEM only (operated for optimized profits) [M€] | $21.3610 \pm 0.0044$ | $22.1238 \pm 0.0040$ (▲3.57% w.r.t. base) |
| (c) On JERM (operated with reserve) [M€] | $25.8388 \pm 0.0046$ | $26.6601 \pm 0.0042$ (▲3.18% w.r.t. base) |
| Expected energy supplied | | |
| (d) On DAEM only (operation for prod. max.) [GWh] | $446.21 \pm 0.03$ | $458.86 \pm 0.03$ (▲2.83% w.r.t. base) |
| (e) On DAEM only (operated for optimized profits) [GWh] | $395.40 \pm 0.03$ | $407.73 \pm 0.03$ (▲3.12% w.r.t. base) |
| (f) On JERM (operated with reserve) [GWh] | $347.78 \pm 0.03$ | $359.62 \pm 0.03$ (▲3.40% w.r.t. base) |
| $AEP_{theory}$ [GWh] | $507.85$ | $520.15$ (▲2.42% w.r.t. base) |

For the base layout, it can be observed in Table 5 that results show the same trends already noticed for 2023: profits on
DAEM only are increased when energy bids are made to maximize profits and not power production. Participating to reserve
leads to yearly profits improved by 20.96%, while it was only 7.82% for 2023. A reason for this better improvement is the




overall lower values of day-ahead prices in 2024, thus giving more opportunities to make profits on reserve markets. Indeed,
allowing participation to reserve markets increases profits significantly when the day-ahead market is less profitable.

The best layout optimized on JERM is the same than the one presented in Tab. 4, i.e. optimized with 2023 data. When operated using 2024 data, the optimized layout lead to higher total profits and supplied energy, in the same order of magnitude than for 2023. These results show that the optimized layout obtained with data from 2023 is still relevant for 2024, even though both year showed dissimilarities in wind distribution and prices.

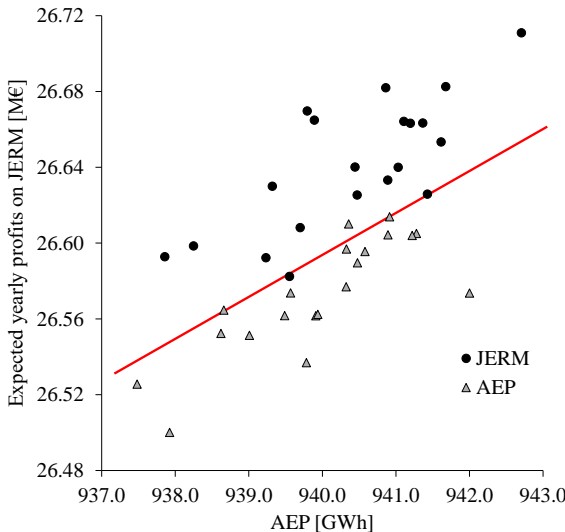

**Figure 12.** Expected yearly profits on JERM in 2024 plotted versus AEP. Each point corresponds to one layout optimized using 2023 data, with the black circles representing layouts optimized with reserve, and the grey triangles are for layouts optimized for AEP maximization.

If we compare again our methodology with the AEP maximization formulation, we observe the same patterns than for 2023. Indeed, when plotting expected yearly profits on JERM in function of AEP, for layouts optimized for profits on JERM and for AEP, we still see that the grey triangles representing layouts for AEP maximization are below the black circles.

## 5   Conclusions

In the forthcoming years, offshore wind farms are expected to have a significant role for restoring frequency balance through
the provision of reserve. Future wind farms should therefore be designed for that purpose. In this paper, a new methodology for WFLO is developed to account for future offshore wind farms participating to secondary upward reserve markets. The objective function aims at maximizing revenues from both day-ahead and reserve markets. It uses stochastic gradient descent for the optimization and probabilistic forecasts of wind power and electricity prices. An inner optimization problem provides the total power contracted on the JERM and the allocation of power to reserve procurement purposes.





When applied on a real-life Belgian test case, results show that yearly profits are expected to increase in a significant way when accounting for participation to reserve markets, while exhibiting a lower supplied energy. This profit augmentation is amplified when the maximum value for reserve bids is increased. Moreover, layouts optimized for profit maximization with reserve markets lead to better yearly profits than when considering day-ahead market only in the objective function. Profits are also higher for the developed methodology than for layouts optimized for AEP maximization, widely used in the literature,

even though the AEP is similar. Finally, the optimized layouts also yield better profits when computed using unseen data. Besides higher revenues, it is critical that wind farms are designed to produce more energy when prices are higher, usually corresponding to periods of electricity shortage. Maximizing production when prices are low or even negative, generally associated with a surplus of generation, leads to spillage of renewable energy.

The perspectives of this work are twofold. First, a better modelling of forecast errors could take into account cross-correlation

between wind, price and activated reserve forecasts (though this would not change the WFLO formulation). Secondly, the impact on blade loads could be relevant to assess the costs versus benefits of providing reserve services. Indeed, wind farms participating in the reserve market should have less fatigue loads, due to a reduced activity (less energy supplied), which would increase the farm lifetime and reduce the operation and maintenance costs of the wind farm components.

*Code and data availability.*   The code and data used in this study are available on Zenodo: https://doi.org/10.5281/zenodo.13946931.

*Author contributions.*   FV, JFT and EDJ formulated the research goals, JQ, PER and THN designed the experiments, JFT, JQ, PER and THN developed the problem formulation, THN performed the simulations, THN and JQ prepared the paper with contributions from all co-authors.

*Competing interests.*   The authors declare no competing interests

*Acknowledgements.*   This research is supported by the Energy Transition Funds project "PhairywinD" organized by the Belgian FPS economy.



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
