# Peer review of "Offshore Wind Farm Layout Optimization Accounting for Participation to Secondary Reserve Markets"

_Wind Energy Science, 2024_

## Author Comment (AC1)

**Response to Referee 1**

The authors would like to thank the reviewer for taking the time and effort necessary to review the first version of the manuscript. We sincerely appreciate all valuable comments and suggestions, which helped us to improve the quality of the manuscript. Our responses to the reviewer's comments are described below in a point-to-point manner. Appropriated changes, suggested by the reviewers, have been introduced into the manuscript (they are highlighted in yellow in the revised version). When the line numbers are provided in this response, they refer to the revisions made in the new manuscript. Please note that the reviewer's comments are repeated in italics and our responses are provided in the bulleted sections of text.

**General comments**

*The paper addresses two topics, namely participation in the reserve market, and layout optimisation for the same. The introduction mentions a third contribution, but since that refers to a case study of the previous two topics, I consider that to be inherent to those two contributions.*

*As the title of the paper implies, the layout optimisation is the main topic, and the contribution to participation in the reserve market is subservient to that. However, in the introduction and the conclusions that part of the study is given an almost equal status. The authors could consider repositioning that part of the study as a means for the layout-optimisation study, rather than as a research subject in itself. This would change the evaluation of the participation in the reserve market (in section 4.1) into a validation of the suitability of the formulation for the purpose of layout optimisation. The exploratory nature of these results could then be a spin-off. However, as a dedicated study into the potential of participation in the reserve market, the formulations may be insufficiently accurate and the validation may be too limited.*

*Having said that, the modelling of participation in the reserve market is convincing for the purpose of layout optimisation. The layout optimisation itself is also convincing, although I have some doubt about the use of expected values for the yearly profit and energy supplied (which may be either an issue of clarification or of method). The authors have chosen useful experiments for comparisons. The comparisons of the experiments could be arranged in a more logical structure, and the interpretation of results seems to be somewhat biased by a presumption of the authors about the benevolence of considering participation in the reserve market in layout optimisation. This is also reflected in one of the main conclusions ("layouts optimized for profit maximization with reserve markets lead to better yearly profits than when considering day-ahead market only in the objective function"), which in my opinion is not supported by the results.*

*In my opinion the four issues that require some more consideration are the role of the study on participation in the reserve market, the use of expected values for profit and supply, the order of comparison of the experiments and the interpretation of results that leads to the main conclusions. Besides these four issues, I think the paper should spend a bit more attention on the size of the reserve capacity market, and the consequential relevance of this study. Having said that, most of my detailed comments are merely suggestions and corrections for readability of the paper. Apart from the potential methodical flaw in the use of expected value for profit and energy supplied, I agree with the entire setup and execution of the body of the research. Although I think that the considerations that I give below can have a major impact on the presentation and conclusions of the research, I think that the actual changes that need to be made are not as major, thanks to the high quality of the modelling, optimisation and experiments.*

- We agree with the reviewer that the main contribution of the paper is the layout optimization problem for a new wind farm, for which we formulate an improved objective function. The latter allows to take into account the participation of future wind farms to reserve markets during the design process of the park. The test case shows interesting findings when we apply our objective function, but it is not a contribution per se.

- The section about the contributions of the paper has therefore been modified and now focuses on the main contribution (P3L71-79)

- We also agree with the reviewer that our work is **not** a dedicated study into the potential of participation in the reserve market. We are fully aware that our formulation to compute expected revenues with reserve provision includes several simplifications, but it is really a first attempt at including reserve markets in the design of future wind farms, and to estimate the impact on the resulting yearly revenues.

- The authors fully acknowledge that the short-term operation of the wind farm used in the study is a simplified representation of reality. As a reminder, we aim at having a long term investment decision: developing a detailed operational formulation for reserve participation of offshore wind farms was not the goal of this paper. In a long-term investment optimization problem, it is very hard to fully include all the complexity and uncertainty related to the short-term operation. In that regard, our study tried to capture the main features of the short-term operation to have a representative, yet simplified, vision of the short-term perspective to properly inform the long-term investment decisions.

- Based on the reviewer's suggestions, in the revised paper, we have included a **new subsection ('Summary of assumptions') to clearly state the hypotheses that we make regarding participation to reserve markets**. We also give a short explanation on how we could address some of these limitations in future work.

- We agree with the reviewer that the order of experiments could be improved in a more logical structure. Therefore, **we followed the proposed order**: comparison of AEP-optimized with base layout, then AEP-optimized with DAEM-optimized, and finally JERM-optimized with DAEM-optimized. While doing so, we have also reinterpreted some the results.

- **The conclusions have been modified** to lower the strength of our claims, focusing on the practical relevance and robustness of JERM-based layout optimization, rather than affirming strict superiority.

- We will address in details the four issues mentioned by the reviewer in the next section about specific comments.

**Specific comments**

**Significance of the reserve capacity market**

*The reserve capacity market is small compared to the day-ahead market. The maximum required capacity of 117 MW at any instant during 2023 is indicative of this. The authors justly argue that this market will grow. However, there is insufficient understanding of how much wind farms will be able to contribute to this market in the future, considering the correlation between their causing the problem (in the day-ahead market) and their ability to provide the solution (in the reserve market). Either way, the conclusion "Future wind farms should therefore be designed for that purpose" (line 460) does certainly not apply to all wind farms, due to the limited size of the market. Although the quoted statement is given in an introductory part of the conclusion, the authors should be careful with such a statement in the conclusions chapter, since this is not sufficiently supported by the outcomes of this research. I think the issue of the size of the current and anticipated future market needs some attention in the introduction and careful formulation of the conclusions.*

- Considering the uncertainties related to the future contribution of wind farms to reserve markets, we agree that we should be more cautious with our affirmation. Therefore, we modified the sentence on P27L550-P28L552 to state the following. If reserve markets grow and the contribution to wind farms to reserve provision becomes important, then future farms should account for those flexibility requirements in their design procedure, even though the limited size of the reserve market might not allow every wind farm to fully participate.

- Regarding the issue of the size of the current and anticipated future market needs, we did a short literature review of the situation in other European countries [1]. In France, the TSO (RTE) has prescribed daily an average of 709 MW of aFRR to the French stakeholders. In the Netherlands, the determined dimensioning minimum of aFRR up was 324 MW in 2023. The Nordic aFRR up capacity market (which covers Eastern Denmark, Sweden, Finland and Norway) has a volume need of 300 MW. However, those needs are expected to increase in the future. For example, in Denmark, for the west bidding zone DK1, the current aFRR up need is 100 MW, but it is expected to reach up to 194 MW in 2035, and 298 MW by 2040 [2], according to the Danish TSO.

- We have added this information in the introduction on P2L37-43.

- The question of how much wind farms will be able to contribute to the reserve market in the future is highly complex. While there could be some moments with a correlation between their contribution to imbalance (in the day-ahead market) and their ability to provide the solution (in the reserve market), other sources of electricity are also a cause for imbalance. Solar energy, sudden changes in load, unavailability of imports from neighbouring countries, ... could also create an imbalance between generation and consumption. Moreover, the contribution of wind farms to imbalance also depends on their penetration in the electricity mix of the considered power system.

- In our test case, we have set a maximum value for reserve bid to 50 MW, which is quite a high value. We argue that it allows us to have an upper level quantification regarding the expected revenues from the reserve markets and the impact on the design. It should be kept in mind that even if the wind farm bid is set to its maximum value, it does not mean that the full bid will be activated, as it depends on the system imbalance and the other flexibility providers in the system.

- For further information, we computed the expected revenues for the base layout of Northwind in 2023 for increasing values (ranging from 2 MW to 50 MW) of $R_{max}$, the maximum value for reserve bid. As represented in Figure 1 below, we can see a linear relationship between expected revenues and $R_{max}$, which shows the high value of flexibility in modern power systems.

**Role of study on participation in reserve market**

*To perform layout optimisation that considers participation in the reserve market, the performance of this participation under a reasonable bidding strategy has been modelled. Several simplifications are made for this. Forecast errors of all forecasted parameters are modelled with independent, gaussian distributions. The paper already mentions several aspects that might be violated by this assumption. E.g., the authors observe*

[Figure]

Figure 1: Expected yearly revenues for the base layout in 2023 for increasing values of $R_{max}$

that "Reserve and regulation prices are characterized by higher volatility, lower mean, more frequent price spikes and a more skewed distribution compared to electric energy prices. Thus modelling their behavior is potentially more challenging" (line 184-185). However, what concerns me most is the neglect of correlations. For instance, "a small prediction error in wind speed can lead to a tremendous need of reserve" (line 291). This indicates a potentially significant correlation between wind speed forecast errors around cut-out wind speed with reserve activation. It is also noted that the optimiser could base bidding on the ratio between day-ahead prices and imbalance fees (line 315-317). However, imbalance fees may be correlated with forecast errors, especially if multiple bidders of wind energy have a systematic error in said forecast, since they use the same or similar weather forecasts.

- The relationship between forecast errors in wind speed and forecast errors on day-ahead prices and imbalance fees is not straightforward. Indeed, if other market players make forecasting errors, but in the opposite direction, errors could compensate and imbalance is not worsened. Hence, it is not certain that there will be a strong correlation between our own forecast mistake (e.g., in wind speed) and the subsequent electricity prices. Moreover, the influence of wind farm forecast errors on the balancing needs also depends on the uncertain penetration of wind energy in the future electricity mix. Overall, we agree that our assumption may potentially be improved, but the solution is clearly not straightforward, especially considering that current available data may not properly reflect the future system conditions.

- Also, we do not believe that all wind parks will all use the same forecasting model (from the same provider) in the future. Indeed, there is a trend to internalize forecast skills within companies, combining in-house capabilities with specialized third-party services to enhance forecasting accuracy and operational efficiency. For example, Iberdrola, a leading wind-power producer, uses advanced machine learning techniques for wind power forecasting. They have collaborated with an AI consulting company for the development of a model able to predict the energy production of wind farms and thus to accurately anticipate wind power production capacity [3]. Vestas' Scipher.Fx uses ensemble forecasting methodology, combines Numerical Weather Prediction data with measurement data, and employs advanced machine learning models to generate accurate power forecasts [4]. Ørsted have developed the wind industry's first uncrewed survey vessel, which uses onboard lidar to accurately predict offshore wind generation [5].

- Therefore, although we are aware of the simplifications we make on forecast errors, we believe that it is still suitable for our application. However, for the sake of transparency, we have added our simplification to the section 'Summary of assumptions' on P10L249 in the revised manuscript.

Another simplification is the replacement of the penalty regulations by a penalty price. The current formulation allows the optimiser to exploit the penalty system, if it just comes at a (monetary) price. The result for the reserve penalty in table 2 shows that this is substantial, compared to the reserve profits. This might indicate that in reality the TSO might already have imposed restrictions on participation to this BSP. (Whether the conditions of the actual regulation are met could be checked from the results a posteriori.) The reserve penalty is implied to be much more substantial for the cases with higher bid limits, thus expectedly

*having a higher number of bid periods with failure to deliver. One can expect that this will only increase in future, weather-dominated systems. As argued above for the potential correlation between imbalance fees and power forecast errors, it can be argued that reserve activation becomes more and more correlated with times of over-forecasting wind power. Since these will also be times where the ability to meet reserve capacity activation is at risk, even if it is prioritised, the risk of failing to meet the capacity test may be underestimated in this study. Furthermore, consistent failure to deliver by wind farms operating in the reserve market may lead to changes in the penalty regulations.*

- We have added this simplification (that the regulations to meet reserve capacity is modelled by a penalty price) in the section 'Summary of assumptions' on P10L250.

- There is a trade-off between having a penalty of infinite value (to represent the expulsion of the market in case of failed availability tests, and in general not being able to provide bidded reserve capacities), and penalties that do not discourage deviations. Penalties that are too low would lead to a bad design, but penalties that are too high would as well. In our case, as stated by the reviewer, reserve penalties are high, which was deliberate to have a trade-off.

- Having a highly detailed operational model was not the objective of this work. Here, we focus on an investment model, which includes making some assumptions for the market formulation. However, the optimized design is "aware" of reserve penalties.

- In the future, if many farms provide reserve services, market and technical constraints might change to account for uncertainties in wind generation. If all wind farms are kicked out of the reserve market, then there might be problems for the provision of reserve. Moreover, we might also expect wind farms to invest in a small battery to avoid penalties for failing to provide the reserve capacity.

- Even though this is not the focus of the paper, we can suggest potential ways to improve on this. For example, Lagrange coefficients can help finding a trade-off for the value of the penalty coefficient. Such coefficient govern how much weight we give to one objective (maximizing revenues on the reserve markets) versus another (avoiding reserve penalties).

*A third simplification is that reserve capacity and reserve activation are aggregated over the bid period. Although there is some inconsistency in the use of power and energy in the paper (see the later technical corrections), the formulations that are used are effectively energy based, aggregating power variations during the bid period to a single value. Although the secondary reserve market allows delayed response, variations of available power of the wind farm and timing of reserve activation within the bid period do matter. An episode of lower power availability could be compensated for the day-ahead market by an episode of higher power in the same bid period. However, it could negatively affect the ability to deliver reserve capacity during this episode of low power in a timely manner. This increases the potential reduced ability to deliver reserve capacity that was discussed above.*

- First, we have corrected the inconsistency in the use of power and energy in the revised paper, we thank the reviewer for noticing this error.

- Second, we agree that there is simplification in our formulation regarding the provision of reserves within a bidding period: we aggregate power variations over a bidding period into a single averaged value. We have added this to our section 'Summary of assumptions' on P10L251-252.

- This simplification is acceptable if power variations within the bidding period are not too severe with respect to the single value. Indeed, Elia, the Belgian TSO, computes the aFRR energy discrepancy with a tolerance band, thus permitting small deviations (15%) from the sent aFRR signal [6].

*The report also refers to the relevance of the study to future, weather-dominated systems, where reserve markets will play a bigger role. I concur with the latter, but the modelling of the reserve market might require significant modifications for such a future system. As argued above, the role and performance of wind farms in the reserve market might change significantly when wind farms become a more important factor in the problems that need to be solved by this market. In the very least, the prices will change (dramatically) for such future systems, possibly accounting for the drawback of decreased reliability of the reserve bids.*

*Because of the doubts that this raises on the accuracy and (untested) validity of the simplified model, I suggest that this aspect of the study is not presented as an inherent contribution (in the introduction) with separate conclusions. Presenting it as a means for the research on layout optimisation, with the associated lower burden of accuracy, seems more appropriate and matches better with the expectations set by the title of the paper. Otherwise, more validation would be required, especially to support the conclusion "results show that yearly profits are expected to increase in a significant way when accounting for participation to reserve markets, while exhibiting a lower supplied energy. This profit augmentation is amplified when the maximum value for reserve bids is increased" (line 4655-467).*

- We agree with the reviewer that considering the involved simplifications, it might be better to present our model for participation to reserve markets as a means for the research on layout optimisation rather than a contribution as such. We have modified our statement of contributions in the introduction to focus on our main contribution: developing a new objective function for the wind farm layout optimization problem that allows, to some extent, to take into account the participation of future wind farms to reserve markets during the design process.

- Also the authors would like to acknowledge that the paper is focused on a long term investment decision (which is informed by the short-term operation of wind parks in energy and reserve markets): developing a detailed operational formulation for reserve participation of offshore wind farms was not the goal of this paper. In a long-term investment optimization relying on short-term operation, it is very hard to fully include all the complexity and uncertainty. This has been clarified in the revised manuscript on P9L240-243.

**Use of expected values for yearly profit and energy supplied**

*On p.4 the use of forecast uncertainty and Monte-Carlo sampling of a set of S forecast errors is explained. Eq. (11) on p.8 articulates how these forecasts are used to optimise bidding. It is clear that this is a probabilistic formulation to deal with the forecast uncertainty. However, it is unclear why and how uncertainty comes into the picture in the yearly profit and energy supplied. The data of realised wind speeds, wind directions, prices, fees, etc. is available, so one would expect the profit and energy supplied to be deterministic, once the bidding is known. In other words, in e.g. Eq. (10) on p.7, the profit can be calculated directly from the realised conditions and operation, rather than determining an expected value for a set of forecast conditions. It is not clear why and how the authors use this stochastic formulation of profit (and energy supplied), leading to values for mean and standard deviation in the tables. Associated places that added to my confusion about this are:*

- The authors understand the confusion and would like to clarify that Problem (11), subject to constraints (12)-(13), mimics the short-term operational optimization of a wind park under forecast uncertainty, using Monte Carlo sampling to account for possible realizations of wind production and prices. The expected profit in Eq. (11) thus corresponds to the "ex-ante evaluation" of the bidding strategy.

- While realized data (wind, prices, etc.) is available in hindsight, we do not recompute an ex-post profit. Therefore, in the training and evaluation of the bidding strategy, we rely on the expected revenue, not the deterministic ex-post value.

- The authors argue that this assumption is reasonable, and enables to simplify the training framework. This has been clearly mentioned in the revised paper on P9L234-238.

- **Indeed, the computation of realized revenues is not the goal here, as one would need a very detailed operational model to do that.**

*p.5, line 134: Perfect forecasts are used to get revenue from the day-ahead market with Eq. (4). For the optimisation of bidding the equivalent of this equation is used with imperfect forecasting, while for the actual revenue one would expect to use the realised situation, rather than a 'perfect forecast'.*

- We apologize for the confusion. Eq. (4) is given with perfect forecasts to simply illustrate how to compute the revenues from the day-ahead market.

- In our framework, we use the equivalent of this equation with imperfect forecasting to emulate the decision-making of a wind park facing operational uncertainties. The resulting ex-ante profit is used as 'loss function' in the gradient-based learning procedure.

*P.7, Eq. (6): The supplied activated reserve uses the wind power forecast, where one would expect the use of the realised wind power, based on the realised wind conditions.*

- This is a consequence of our assumption to use an 'ex-ante operational profit' in the loss function.

*p.9, Eq(14): Here the objective function for layout optimisation is expressed as an expected value, using the samples of forecast errors that where generated for the optimisation of the bidding strategy. Nevertheless, on line 222 it is stated that the objective is to maximise profit, and not the expected value of profit.*

- This is a text mistake, as the objective is to maximize **expected** revenues. We have corrected the typo.

*p.9, line 239: This mentions that results are computed as expected yearly profits and supplied energy, instead of deterministic yearly profits and supplied energy.*

*p.14, caption of table 1 (and later tabulated results): This table shows the expected values and explains in the caption how the mean and certainty interval are associated with the sample set S of forecast errors. Again, this association is understandable for the profits and available power used in the optimisation of bidding (related to p.8, Eq. (11)), but is not what one expects to use for the realised profits and energy supplied.*

*The use of these expected values should be better argued and explained, or (more likely) these parameters should be treated as deterministic parameters.*

- We would like to clarify that we merely use historical data to have realistic (and somehow correlated) values of wind and electricity prices. They are only meant to be leveraged to provide base values for our simplified forecast approach.

- The computation of realized revenues is not the goal here, as one would need a very detailed operational model to do that. When we refer to revenues in 2023, what we really mean is the expected revenues if wind and prices behaved in a similar way than they did during that year.

- As a summary, we want to study many possible different futures, which is why we used expected profits (several possible realizations of the future). Computing a realized profit (using historical data) could be done but the signification would be different for the obtained revenues. Doing so would imply that if we look back, this is the profit that we would have got if we had chosen this specific path of realization.

**Order of comparisons of experiments**

*In section 4.2 the performance of a layout optimised for JERM is compared to that of the base layout. The risk of this approach is that improved performance of the JERM-optimised layout is assigned disproportionately to JERM-optimisation, whereas a majority of the improvement could have been reached with other optimisations as well. Indeed, the discussion of the results immediately zooms in on the 3.10% improvement on JERM, while the significance of the 3.58% increase for the simplest participation on DAEM only is ignored. That increase indicates that much of the improved performance could be assigned to AEP improvement of the farm, rather than to JERM optimisation specifically. The AEP improvement is mentioned and analysed, but without consideration of the meaning of that for the association of JERM-optimisation with the total improvement of 3.10%.*

*The authors later compare JERM optimisation with AEP and DAEM optimisation. Indeed, the expected logical order of performance on JERM is recognised to be: base layout, AEP-optimised, DAEM-optimised, JERM-optimised. However, by first making the jump from base to JERM-optimised and then do a backward analysis to the intermediate optimisation options, the perception and interpretation of the results is biased by the first indications. In my opinion, the contribution to JERM (and DAEM) performance of different types of optimisation would have been much clearer when the experiments were done and shown in the order of expected optimality, comparing the improvements one step at the time: base layout to AEP-optimised, AEP-optimised to DAEM-optimised, DAEM-optimised to JERM-optimised.*

*As an extension of the chosen approach, the performance to the unseen 2024 data in section 4.5 focused on a comparison of the base layout with the JERM-optimised layout. Also here the robustness of JERM optimisation is obscured by the large improvement in AEP (visible in the large improvement of performance on simple DAEM-only operation). I think the specific merits of JERM optimisation would become clearer in a comparison of DAEM- and JERM-optimised layouts operating in 2023 and 2024 markets. Comparison of AEP- and DAEM-optimised layouts for the same could serve as a baseline, to assess which differences can be attributed to robustness to market conditions (DAEM- versus JERM-optimised) and which to robustness to wind conditions (AEP- versus DAEM-optimised).*

- The authors understand the point of view of the reviewer. In the first version of the paper, we decided to start by our most meaningful contribution, then we added benchmarks and comparisons with other methodologies. In this way, the added value of the methodology was discussed at the start of the case study.

- To avoid any misleading into the benefits of JERM (with respect to simply participating to DAEM), **we have modified the structure of the case study**. We explain the order of experiments at the end of the Test case section, on P13L298-302.

- Hence, the specific merits of JERM optimisation are clearly nuanced, by firstly showing the benefits of going from the base layout to AEP-optimised, then the AEP-optimized to DAEM-optimized, and finally from DAEM-optimized to JERM-optimized.

- For the performance on unseen 2024 data, we have also modified the comparisons to follow the same logical structure than mentioned above.

**Conclusions and interpretation of results**

*I will first reflect on some interpretations of the results and then give my own interpretation. Subsequently, I'll address how this might affect the conclusions.*

*I already addressed the bias caused by the interpretation of results in section 4.2. Therefore, I continue with the analysis of figure 8 on p.19. I will continue to use 'DAEM-optimised' for what is called 'Optimized without reserve' in the figure, and 'JERM-optimised' for what is called 'Optimized with reserve'. Each point in the graph is an optimised layout. The scatter indicates the stochastic nature of the optimisation. The width of the scatter, when compared to the difference between the DAEM-optimised and JERM-optimised points indicates that no direct comparison can be made between any best performing layouts. That would put more emphasis on 'luck' of drawing a good sample from the layouts, rather than on the difference between the two optimisation types. Somewhat in line with the discussion of the authors of figure 9 on p.20, the fairest comparison in performance on JERM seems to be to draw a diagonal fit through all DAEM-optimised layouts and a fit through all JERM-optimised layouts, and to compare those. These two fits would be almost the same. If JERM-optimised layouts would consistently perform better on JERM than DAEM-optimised layouts, one would expect the fit to JERM-optimised layouts to lie higher than that of DAEM-optimised layouts: Any layout that achieves a certain performance on DAEM, irrespective of how they were optimised, should achieve a better performance on JERM if it was JERM-optimised. A similar argument is applied by the authors in the discussion of figure 9 on p.20, based on the observation that the scatter of circles lies upward of the scatter of triangles. On p.20, the authors associate the downward shift of the AEP-optimised layouts compared to the JERM-optimised layouts with the performance characteristics of AEP optimisation. It seems inconsistent to then not associate the lack of a downward/upward shift with equal performance for DAEM and JERM optimisation. In my opinion the only conclusion that is supported by the results in figure 8 is therefore that JERM-optimised layouts do not perform better than DAEM-optimised layouts, neither on DAEM, nor on JERM.*

*Although the authors discuss the general shift between the points in figure 9, the magnitude and significance of this shift is not addressed. Two diagonal fits between the two sets of points would have a vertical shift of about 0.1 MEuro, which corresponds with less than 0.15%. Even between the highest JERM-optimised and highest AEP-optimised points the difference is only about 0.2 MEuro: less than 0.3%. Figure 12 indicates an upward shift of about 0.08 MEuro for unseen data of 2024, corresponding to about 0.3%. These percentages are most indicative of the performance of DAEM optimisation over AEP optimisation.*

*In the discussion of figure 8 the authors zoom in on the performance of the two two right-most points for JERM-optimised farms. Liekwise, on p.20 they focus on the right-most point in figure 9, addressing that it performs better on AEP than the (best) AEP optimisation. As argued above about the scatter of the results, these results seem inconclusive as to the inherent superiority of the two right-most points in figure 8 and the one right-most point in figure 9, as opposed to their 'lucky sampling'. The authors argue that JERM optimisation might be slightly more likely to find solutions with high optimality (for AEP, DAEM, as well as JERM performance), due to better gradients (line 396-398 and line 412-413). However, this doesn't confirm the significance of using JERM as an objective over using AEP or DAEM, but rather the significance of how either objective is formalised and how the problem is solved. In other words, it doesn't mean that these layouts are better optimised for JERM per se.*

- The authors do not fully agree with the statement that "results seem inconclusive as to the inherent superiority of the two right-most points in figure 8 and the one right-most point in figure 9." Indeed, not only those points show that, over several runs of the training algorithm, the maximum profits can be reaped by the JERM-optimized layout, the outcomes also reveal that these optimized layouts for JERM are better in average. The whole distribution of profits (over experiments) is improved, not only extreme points.

- We fully agree with the reviewer that the stochastic nature of the layout optimization introduces variability in the results. As such, we focus on the performance trends across diverse runs. We argue that our analysis includes both the best cases (which is, as such, very important) and the average performance.

- Our intent in discussing the shifts in Figures 8 and 9 was to highlight general tendencies rather than absolute gains from JERM optimization. While we agree with the reviewer that the observed shifts (0.1–0.3%) are small in magnitude, they are consistent across multiple optimization runs. This suggests that JERM-optimized layouts are at least as effective as DAEM-optimized layouts, and even offer slight improvements.

- We appreciate the reviewer's observation regarding the role of gradient quality in optimization convergence. Indeed, our statement that JERM formulations can lead to more navigable optimization landscapes reflects that the structure of the objective function may help guide the search toward better-performing regions. However, we agree that this should not be interpreted as inherent superiority of the objective.

- We also would like to inform the reviewer that there was a mistake in Table 4 for the results of the layout optimized on JERM. Indeed, the values were too low, and did not correspond with the results seen in Fig. 8 and 9. This has been corrected in the revised version.

*Considering the above, I propose a re-interpretation of the results along the lines of the proposed reordering of the experiments. Since I don't have all (intermediate) results, my interpretation will be rough and based on ball-park figures:*

*- From base layout to AEP-optimised: Improvement of performance on DAEM and JERM of about 3%. This is based on the results in Table 4, and the subsequent minimal contributions that I identify for the other optimisation improvements.*

*- From AEP-optimised to DAEM-optimised: Improvement on DAEM and JERM of about 0.15-0.3%. For the improvement on JERM, I base this directly on figures 9 and 12, as discussed above. The improvement on DAEM would be similar, due to the absence of improvement between JERM and DAEM optimisation.*

*- From DAEM-optimised to JERM-optimised: No observable improvement. This is based on my discussion of figure 8 above.*

*This interpretation of the results identifies the improvement attained by layout optimisation for AEP as the main cause of the improvements seen in JERM optimisation, with a small contribution of optimisation for DAEM. It identifies no improvement of JERM optimisation over DAEM optimisation. I fully agree with the underlying mechanism that the authors discuss for AEP optimisation, as well as for the mechanism of changing weights for wind-direction sectors in case of DAEM optimisation. However, I don't think the results support any conclusion about the significance of and need for JERM optimisation, such as "this highlights the importance of accounting for participation of wind farms to reserve markets in the layout optimization*

*process" (line 393-394) and "layouts optimized for profit maximization with reserve markets lead to better yearly profits than when considering day-ahead market only in the objective function" (line 467-468). I think the authors should reflect on such conclusions and suggestions of mechanisms to support them considering the previous discussion.*

- We thank the reviewer for this detailed re-interpretation. We agree that the major performance gains are partly due to the initial AEP-based optimization, with smaller but consistent improvements from DAEM optimization. Then, while the incremental gain from JERM over DAEM appears modest, we emphasize that:

  - The inclusion of reserves in the objective better reflects real-world market conditions, and ensures that layouts are evaluated under more comprehensive profitability criteria—not just energy yield.
  - Even small profit differences ($\sim$0.1–0.3%) are economically meaningful at scale, and JERM-optimized layouts consistently perform at least as well as, and occasionally better than, DAEM-optimized ones.
  - We have revised the wording of our conclusions to lower the strength of our claims, focusing on the practical relevance and robustness of JERM-based layout optimization, rather than affirming strict superiority.

**Technical and textual suggestions and corrections**

- Syntax and grammar errors have been corrected, and minor text improvements have been integrated in the revised paper.

*Please be more precise with the distinction between power and energy. Where needed, add '\*delta_t' for the duration of the time step, to get from power to energy. As an example, revenues in equation 4 are derived from power times price, where prices are given e.g. in figure 4 in Euro/MWh. Figure 4 also gives price for reserve capacity in Euro/MWh (instead of Euro/MW), for which it is not clear whether or not that is consistent. Please also take this into account for reserve activation $R*k_a$, which should be in terms of energy and not power (see also figure 1). The true implementation of power and energy in the model seems correct, since the revenues for DAEM only correspond with an estimate of them with a reasonable capacity factor for the wind farm.*

- Indeed, all prices are given in €/MWh and should be used with energy, and not power (if power is used, it should be multiplied by the duration of the timestep $\Delta t$). We have corrected this mistake in Eq. 4 and throughout the revised manuscript (equations 7, 8, 9, 10, 11, 14).

*Please be more precise with the use of the term profit. In many places, such as in table 1, 'profit' is used, where '(net) revenue' is meant, since costs for the wind farm are not accounted for. For convenience, I copied the use of 'profit' in my comments above, also where this is not correct.*

- We agree with the reviewer that since we do not account for wind farm costs, we compute net revenues with our formulation, and not profits. We have replaced the term 'profits' with 'revenues' throughout the revised version of the paper.

- We have also added a short sentence on P8L208-209 emphasizing that **net** revenues are computed with our new objective function.

*It could be helpful if chapter 2 already stated for which parameters data is used. That closes the set of equations. (So, for wind speed and direction, day-ahead prices, reserve market capacity and activation prices, imbalance fees, reserve market activation level, other bids in the reserve market, ... ?) As specified in a few comments below, for some of these parameters the text causes confusion as to whether these parameters are modelled, or whether (only) their forecast errors are modelled.*

- Forecasts of wind speed, wind direction, day-ahead prices, reserve capacity and activation prices, and activated reserve volumes are all modelled, based on historical data to which we add a forecast error that we model using a normal distribution.

- We added that information on P10L246-248.

*2, line 23: Please rephrase for readability.*

- We replaced that sentence on P2L22-25 by the following.

- Moreover, it has been proven that modern wind turbines with variable rotation speed have intrinsic fast ramping down and ramping up capabilities, which can be effectively used to provide ancillary services. Ramping down is virtually done at no cost (if prices for down reserve are negative), and ramping up is subject to the availability of wind power.

*4, line 115-116: I propose to scratch '(normally not known by the wind farm operator)'. This is never known, by anyone.*

- Indeed, forecast is by definition an estimation of the future and is not know before the actual realization. We removed that part of the sentence on P5L125-126.

*Forecast error modelling:*
*Eq. (3): The modelling error associated with the wind farm model is not explained, nor are the values for its mean and standard deviation given.*

- The modelling error for wind farm power depends on the wind farm model that is used to convert wind speed and wind direction to wind power production.

- In this paper, PyWake is the wind farm model, and we use modelling errors found in the literature where the analytical models of Pywake are compared with SCADA data [7]. The mean error is zero (as PyWake is shown to underestimate or surestimate the power production depending on the wind farm), and the standard deviation is set to 3%.

- We have added this information in the test case section where we present PyWake as our wind farm model, on P13L310-312.

*Line 138: Can this sentence please be corrected or clarified. It seems that the distribution of forecast error is meant, but it states 'of day-ahead electricity prices' (for which a zero mean makes no sense). Can be articulated of what the percentage is taken?*

- Indeed, we meant a zero-mean for the **distribution of forecast errors** of day-ahead prices, not for the distribution of day-ahead prices. We have corrected this mistake on P6L151.

- It should be noted that we use a zero-mean for the distribution of forecast errors since when a forecast model is trained for mean square error minimization, we "force" MSE towards zero.

*Line 184-186: It is unclear whether these sentences are about modelling of prices, or (as would be expected) modelling of their forecast errors. Could that be clarified? Does 'regulation prices' mean 'imbalances fees', for which the (forecast) model has not been addressed elsewhere? Can the mean and standard deviations used for forecast errors of reserve capacity prices, reserve activation prices and imbalance fees be given? The current last sentence of this paragraph is open-ended.*

- These sentences refer to forecasting reserve prices, as we do not model these values in our work. We have clarified this sentence on P8L200.

- We still set a zero mean value for the distribution of forecast errors. For the standard deviation, no clear value can be found in the literature. However, we assume that the forecast inaccuracy is expected to be higher than for day-ahead prices, so we set a higher value of 10%. We have added this information on P8L201-203.

- Yes, the regulation fees are what me called elsewhere the imbalance prices. To avoid confusion, we have replaced the former by the latter in the text.

*6, line 149: Could be specified to which assumption is referred here? The previous text only describes certain mechanisms and the choice of only providing upward reserve regulation; it gives no assumption.*

- The assumption refers to the fact that prices for downward reserve are usually negative (the BSP pays the TSO), and positive prices (the TSO pays the BSP) only occur in specific conditions. We have clarified this in the text on P6L161-162.

*6, line 158-160: Could you clarify how the distribution of reserve activation amongst multiple bidders is modelled? Is data about bids in 2023 available and used? Furthermore, could be clarified how is determined if or how much of the reserve capacity bid is won, in case of multiple players and/or limited reserve need? The current formulation (Eq. (14)) implies that any bid is always won in full.*

- Since reserve capacity bids are awarded based on the bidded price, determining how many MWh are awarded to our wind farm would involve knowing the behaviour of other market players.

- In future work, what could be done is to simulate several wind farms, make them all participle in the reserve market, then award reserve quantities based on the bidded price. This is a rather complex process, possibly involving game theory since the market players might behave differently based on the bidded price of other players. Moreover, other players operating different technologies (e.g., batteries, demand side response, ...) could behave in a different way than wind farm owners.

- Considering the high complexities involved in the modelling of distribution of reserve activation, we made some simplifications: historical data regarding the total activated reserve is used, and is divided by the required volume of aFRR reserves in Belgium (117 MW in 2023 and 2024). This gives us scenarios of reserve activation, denoted by $\kappa_{k,t}^a$ in equations. We then apply this pro-rata to determine how many MWh are awarded to the wind farm. Therefore, in our formulation, we do not assume that any bid is always won in full: the quantity awarded by the TSO depends on the system needs.

- We have added this information on P10L253-257.

*7, line 171-172: It is unclear how the availability tests for reserve capacity is implemented. In Eq. (10) and (11) the penalty of this test (= Eq. (8)) appears at every time step. Does this mean that the test is effectively done every time step, or is delta$_R$ set equal to zero at all but 12 (random) time steps? Does the algorithm for optimising operation (Eq. (11)) in any way account for a probability that this test is performed in that time step, or is it assumed that the test will be performed in that particular time step with 100% certainty?*

- We assumed that the test will be performed in each timestep with 100% certainty. Indeed, since the technical penalty (reduction of the bid upper limit in reserve markets) incurred by the wind farm for failing consecutive tests is not directly modelled, adding a probability to the availability tests led to unrealistic reserve bids and expected revenues.

- We have added this assumption on P10L258-260.

*7, Eq. (9), (10), and several other related equations: Consider not to replace gamma$_a$ by 1.3, since other values are also not substituted. The superscript c has been incidentally dropped from gamma in Eq. (10) and several other equations.*

- We have modified equations (9), (10), and following equations to make $gamma_a$ appear instead of its value. We have also added the superscript c where needed in equations (10), (11) and (14).

*8, line 200-201: Please formulate more clearly what is meant by the 117 MW. Is that the highest needed reserve capacity in that year?*

- The 117 MW correspond to the volume of aFRR to procure at all times by Elia in 2023. This value is updated every year and results from an optimisation based on costs, using a probabilistic method based on a time series of two years of expected variations between quarter-hours of system imbalances. It should be noted that this corresponds to the total volume of reserve **capacity** bids that the TSO should award for every timestep. The actual activation of reserve depends on the system imbalance. This has been specified on P12L290-292.

- Article 32(1) of Commission Regulation (EU)2017/2195 of 23 November 2017 establishing a guideline on electricity balancing states that "all TSOs of the LFC (load-frequency control) block shall regularly and at least once a year review and define the reserve capacity requirements for the LFC block or scheduling areas of the LFC block pursuant to dimensioning rules as referred in Articles 127, 157 and 160 of Regulation (EU) 2017/1485. Each TSO shall perform an analysis on optimal provision of reserve capacity aiming at minimisation of costs associated with the provision of reserve capacity.

*8, line 207: Can you clarify to which approach 'this approach' refers? This could be the approach of Soares et al., or it could be the approach that is presented in the current manuscript. If it is the latter, then the last sentence about submission steps of 1 MW doesn't seem to correspond with the given formulation.*

- In that sentence, we refer to the approach of Soares et al., in which the energy and reserve share can be adjusted at each stage. It means that the allocation of energy and reserve during the first stage (day-ahead bidding) can be different from the allocation of the second stage (the actual delivery of power). For example, for a given timestep, let us consider that a wind farm operator forecasts a production of wind power of 40 MW, and allocates 30 MW to the day-ahead energy market (i.e., 3/4 of the wind generation), and 10 MW (1/4) for reserve. If the actual production of wind at delivery time is 35 MW and reserve penalties are high, the wind farm owner does not have to keep the 3/4-1/4 ratio, and can decide to provide 10 MW for reserve activation and 25 MW for the energy market (and thus incur imbalance fees for the day-ahead market).

- We have clarified this on P9L228-229.

- The steps of 1 MW are merely a practical consideration set by the TSO Elia in terms and conditions of a balancing service provider contract.

*9, line 217: smaller than or equal to > larger than or equal to.*

- We corrected the typo.

*9, line 218-221: Eq. (15) keeps the turbines within the boundaries of a square wind farm area. This is not what is used in the case study. Could this be made consistent?*

- Indeed, the constraint of Eq. (15) is not applicable in our test case where the farm boundaries are a polygon. We rectified Eq. (15) on P10L265 with the following:

$$\boldsymbol{x}, \boldsymbol{y} \subset B \tag{1}$$

  where $B \subset \Re^2$ is a closed region in which to place turbines (its edges are the farm boundaries).

*11, line 254-255: Consider making explicit that the data is nevertheless used to optimise bidding for that year.*

- We have added this information on P13L304-305 with the following sentence.

- "Those data were not seen during the layout optimization process but are nevertheless used to optimize bidding for that year."

*11, line 264: Can be clarified what is meant by this sentence? This seems more like an explanation of a multi-start optimisation to improve optimality than an improvement of statistical significance. That also aligns with later statements about results being given for the best performing layout (p.16, line 354-355). In case statistically significant is indeed meant: of which stochastic output?*

- Indeed, since we later focus on the best performing layout, we do not use several initial random conditions for statistical significance. The reason for those different conditions is to avoid lucky/unlucky sampling pitfalls during the SGD optimization.

- We have corrected the mistake in the text on P13L316.

*16, Table 3: Much of this table is a repetition of table 2. Consider whether table 2 can be removed, by using table 3 differently.*

- We have merged the two tables to avoid repetitions.

*24, line 472: Please rephrase 'periods of electricity shortage'. Belgium rarely (if ever) has electricity shortage.*

- Indeed, Belgium does not face periods of electricity shortage. However, periods of low electricity production can genuinely happen. We rephrased 'periods of electricity shortage' by 'periods of low electricity production' in P28L568 of the revised version.

**References**

[1] ENTSO-E, "Balancing report 2024," tech. rep., ENTSO-E, 2024.

[2] Energinet, "Outlook for ancillary services 2023-2040," tech. rep., Energinet, 2023.

[3] Whitebox, "Wind Energy Forecasting - Iberdrola: Development of a model to predict the energy production of wind farms, optimizing their efficiency and profitability." `https://www.whiteboxml.com/en/casos-de-exito/prediccion-de-la-energia-eolica`. Accessed: April 2025.

[4] Vestas, "Scipher.Fx Power Forecasting." `https://www.vestas.com/en/energy-solutions/service/digital-services/scipher/ScipherFx`. Accessed: April 2025.

[5] Ørsted, "Innovation report - Harnessing innovation to create the green energy systems of tomorrow," tech. rep., Ørsted, 2024.

[6] Elia, "Terms and Conditions for balancing service providers for automatic Frequency Restoration Reserve (aFRR)," tech. rep., Elia, 2022.

[7] D. Van Binsbergen, P.-J. Daems, T. Verstraeten, A. Nejad, and J. Helsen, "Performance comparison of analytical wake models calibrated on a large offshore wind cluster," in *Journal of Physics: Conference Series*, vol. 2767, IOP Publishing, 2024.

---

## Author Comment (AC2)

**Response to Referee 2**

The authors would like to thank the reviewer for taking the time and effort necessary to review the first version of the manuscript. We sincerely appreciate all valuable comments and suggestions, which helped us to improve the quality of the manuscript. Our responses to the reviewer's comments are described below in a point-to-point manner. Appropriated changes, suggested by the reviewers, have been introduced into the manuscript (they are highlighted in yellow in the revised version). When the line numbers are provided in this response, they refer to the revisions made in the new manuscript. Please note that the reviewer's comments are repeated in italics and our responses are provided in the bulleted sections of text.

**General comment**

The paper was clearly written and did a good job in exploring the topic. The paper claims 3 areas of contribution. I didn't find anything novel about the wind farm layout optimization, so I think those contributions are overstated. I see one contribution, the first one regarding the formulation of a new objective based on participation in reserve markets.

- We agree with the reviewer that the main contribution of the paper is the formulation of a new objective function for the wind farm layout optimization problem. The latter allows to take into account the participation of future wind farms to reserve markets during the design process. The test case shows interesting findings when we apply our objective function, but it is not a contribution per se.
- The section about the contributions of the paper has been modified and now focuses on the main contribution (P3L71-79)

To the best of the authors' knowledge, this is the first paper that presents a wind farm layout optimization that accounts for the participation to reserve markets in the revenue objective function. Therefore, the main contribution of this paper is the formulation of a new objective function for the wind farm layout optimization problem. The latter allows to take into account the participation to reserve markets during the design process. The new objective function aims at maximizing expected yearly revenues of a wind farm participating in both day-ahead and secondary upward reserve markets. It allows to compute the optimal offering in both markets, reserve allocation strategy, and subsequent expected revenues. The new objective function considers the uncertainty in forecasts of wind power, electricity prices and activated reserve volumes. The estimated penalties and balancing costs for failing to provide energy and reserve are also taken into account. The study is conducted for the Belgian system using existing market rules. However, although this system has some peculiarities, the main methodology could be applied in other systems with minor modifications.

**Main concerns**

Some of the results suggest that the sample sizes are too small (for example the best AEP design does not come from AEP optimization). Also, there is no real evidence to claim that one function "has better gradients" than the other from one data point (again just looks to be a small sample size problem for a problem that is well known to be multimodal).

- We agree with the reviewer that it is worth noticing that the best-performing AEP layout is not the one obtained by directly optimizing the AEP objective.
- Yet, as explained in the first version of the paper, this can be explained by the fact that gradient-based optimization may converge to better solutions when guided by more comprehensive objectives (e.g., JERM or DAEM), which offer smoother and more informative gradients. These richer objectives may implicitly regularize the search process, helping avoid poor local minima and yielding layouts that are not only robust in market performance but also superior in raw energy yield.
- The authors acknowledge the non-determinism of the optimization process, but do not believe that it is a problem of sample size. Indeed, the number of sampled timesteps for each SGD iteration is quite extensive, for many values of K \* T. This is evidenced by the convergence of mean performance across independent runs and the low variance in key indicators such as AEP and revenues. To support this, we have verified that increasing the number of samples or optimization iterations consistently leads to the same observations.
- Overall, it should be noted that we do not make an indisputable claim regarding the better gradient, as we merely try to offer some plausible explanations for this.
- Indeed, if the AEP objective function has a poorly conditioned landscape (e.g., sharp ridges, flat regions), gradient descent might struggle to find high-quality optima.
- More comprehensive objectives (like DAEM or JERM, which combine multiple aspects such as price signals and reserve activation) may offer a better exploration of the solution space and produce better gradients, smoother curvature, and more informative updates, which can guide the optimization toward layouts that are superior across several criteria, including AEP.
- Optimizing a richer objective may act as a form of regularization, preventing overfitting to narrow aspects of performance (e.g., maximizing AEP in a single direction). This broader objective may lead to more balanced layouts, which incidentally perform better even on simpler metrics like AEP.
- However, to avoid confusing the reader, we removed our possible explanation of better gradients. We now state on P24L505-506 that this result (the best AEP design not coming from AEP optimization) is quite surprising and should be further investigated in future work.

The difference between AEP and JERM optimization was minimal ( $\approx 0.1\%$ ) between the best in each category. That type of difference is much smaller than the uncertainties in both the energy and cost metrics, which also makes it hard to make strong claims on improvements.

- The mean yearly expected profits for JERM optimizations is 71.8956 ± 0.105 M€, while it is 71.7638 ± 0.106 M€ for AEP optimizations. The mean absolute difference is 0.1318 M€, i.e., 0.18%, thus increased revenues of 2.6 M€ over the farm lifetime. We have added these results in a paragraph on P23L495-499.
- We agree that the scatterplot of Fig. 9 does not convey this information, and we have added a boxplot on P23 to show summarized results. We believe that while the improvement in expected profits is not as strong as when comparing the layout with optimized design, it is still higher than the uncertainties on cost metrics.
- However, for AEP, we do agree with the reviewer that the improvement in AEP is less noticeable, and it is smaller than the uncertainties in energy metrics. Therefore, we cannot straightforwardly claim that JERM optimizations give better AEPs than AEP optimizations. We added this observation in P23L499-501.

**Minor comments**

In the abstract it would be clearer to specifically state how much higher the profits are for the new methodology when compared to just optimizing with AEP ("Profits are also higher for the new methodology than when using the maximization of annual energy production, widely used in the literature, as objective function.")

• Indeed, this could improve clarity: we have added this information in the abstract of the revised version (P1L11).

Line 39: "This does not allow to capture the variation of day-ahead and reserve prices with wind speed and wind direction." maybe revise this sentence to "This does not capture the variation..."

• The sentence has been revised to what the reviewer suggested.

Line 73: "This allows to obtain rather accurate results in a reasonable computation time." Change to something like this - "This approach enables accurate results with reasonable computation time."

• The sentence has been revised to what the reviewer suggested.

Line 115: Q. How then can the power day ahead be predicted if the operator can't predict the day ahead wind forecast?

- This sentence states that the actual realization of wind is not know by the operator when making power bids. Therefore, the operator should first forecast wind speed (the day-ahead prediction of wind speed is widely studied in the literature) and wind direction, then obtain the corresponding wind power forecast by converting this wind information to power using a wind power model. In our paper, the latter is PyWake.
- We clarified this sentence in P5L134-136.

Line 270: "Prices for reserve capacity and reserve activation, as well as activated upward aFRR reserve volumes are were provided

• We have corrected this typo.

For the paragraph starting at line 415 it would be helpful to better quantify how much better the JERM optimization profit is than the AEP optimization over the range of AEPs.

• As explained in our response to the second main concern, the mean yearly expected profits for JERM optimizations is 71.8956 ± 0.105 M€, while it is 71.7638 ± 0.106 M€ for AEP optimizations. The mean absolute difference is 0.1318 M€, i.e., 0.18%, thus increased revenues of 2.6 M€ over the farm lifetime. We have added these results in a paragraph on P23L495-499.

Line 233 - bee too costly

• We have corrected this typo.

Different values of K \* T is said multiple times but is not very clear.

- The choice of K, the number of days, and T, the number of timesteps in a day, should be a tradeoff between accuracy and computational time. Indeed, increasing values of K and T, i.e., increasing the number of samples for the computation of the total expected profit, allows to encompass more situations at each optimization step. However, this also has an impact on the computational burden.
- Therefore, we ran optimizations for K \* T ranging from 20 to 150. We have added this information in P13L315-316.
- It should be noted that we noticed that at some point, further increasing K \* T did not provide significant improvement in expected profits and AEP, notably with regard to the marked increase in computational time and resources.